# TRAP: Targeted Redirecting of Agentic Preferences

**Hangoo Kang**[*1]**, Jehyeok Yeon**[*1]**,Gagandeep Singh**[1]
[1]University of Illinois Urbana-Champaign
{hangook2,jehyeok2,ggnds}@illinois.edu

## Abstract

Autonomous agentic AI systems powered by vision-language models (VLMs) are rapidly advancing toward real-world deployment, yet their cross-modal reasoning capabilities introduce new attack surfaces for adversarial manipulation that exploit semantic reasoning across modalities. Existing adversarial attacks typically rely on visible pixel perturbations or require privileged model or environment access, making them impractical for stealthy, real-world exploitation. We introduce TRAP, a novel generative adversarial framework that manipulates the agent's decision-making using diffusion-based semantic injections into the vision-language embedding space. Our method combines negative prompt–based degradation with positive semantic optimization, guided by a Siamese semantic network and layout-aware spatial masking. Without requiring access to model internals, TRAP produces visually natural images yet induces consistent selection biases in agentic AI systems. We evaluate TRAP on the Microsoft Common Objects in Context (COCO) dataset, building multi-candidate decision scenarios. Across these scenarios, TRAP consistently induces decision-level preference redirection on leading models, including LLaVA-34B, Gemma3, GPT-4o, and Mistral-3.2, significantly outperforming existing baselines such as SPSA, Bandit, and standard diffusion approaches. These findings expose a critical, generalized vulnerability: autonomous agents can be consistently misled through visually subtle, semantically-guided cross-modal manipulations. Overall, our results show the need for defense strategies beyond pixel-level robustness to address semantic vulnerabilities in cross-modal decision-making. The code for TRAP is accessible on GitHub at `https://github.com/uiuc-focal-lab/TRAP`.

## 1 Introduction

Vision-Language Models (VLMs) and autonomous agentic AI systems have significantly advanced the capability of machines to navigate and interpret open-world environments [Radford et al., 2021, Li et al., 2022a, Alayrac et al., 2022]. However, these powerful multimodal systems also introduce new vulnerabilities, particularly through adversarial manipulations that exploit their integrated visual-textual perception [Zhou et al., 2023, Moosavi-Dezfooli et al., 2016b]. A critical emerging threat is cross-modal prompt injection, in which adversaries embed misleading semantic cues in one modality (e.g., an image) to influence the interpretation and decision making of a model in another modality (e.g., language understanding) [Liu et al., 2023c]. Unlike traditional unimodal adversarial attacks that primarily perturb pixels or text unnoticeably [Goodfellow et al., 2015, Uesato et al., 2018, Madry et al., 2019], these cross-modal attacks leverage semantic shifts, misleading autonomous agents without triggering human suspicion.

Fully autonomous agents, such as GUI agents that navigate web interfaces without human oversight, are particularly susceptible to adversarial manipulations. Recent work has shown that visual-language agents can be jailbroken by adversarial environments, leading to unintended and potentially harmful

---

*Equal contribution

actions [Liao et al., 2025, Zhang et al., 2024b]. For example, a malicious pop-up or UI component could trick an agent into clicking harmful links or executing unauthorized tasks, without human intervention [Wu et al., 2024]. This highlights a critical safety flaw: such agents inherently trust their perceptual inputs, making them highly vulnerable to subtle semantic perturbations [Li et al., 2024].

In this paper, we introduce TRAP, a novel adversarial framework explicitly designed to exploit agentic systems' vulnerabilities through semantic injection using diffusion models. Our approach leverages the generative system of Stable Diffusion Rombach et al. [2022b] in combination with CLIP embeddings to create realistic adversarial images that subtly mislead an agentic AI system's decision.

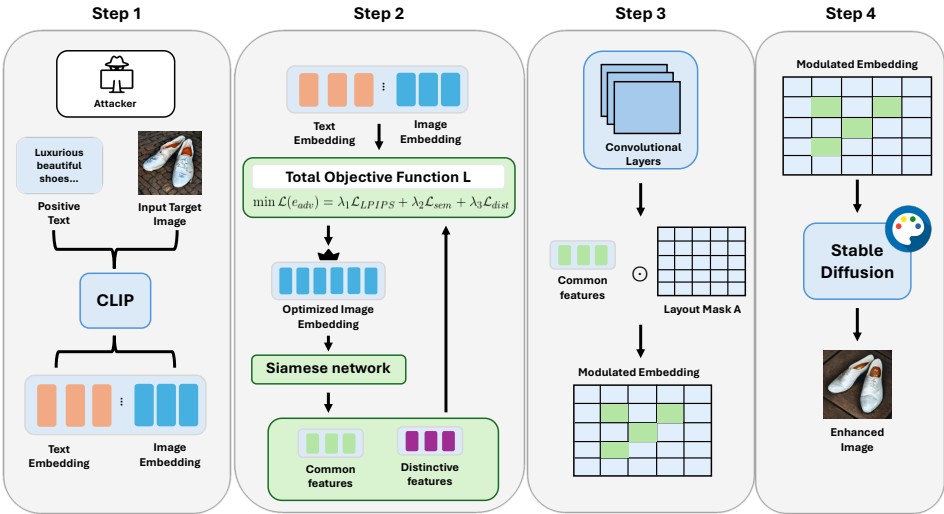

Figure 1: Overview of the TRAP adversarial embedding optimization framework.

TRAP operates in four stages (Fig. 1). First, we extract CLIP embeddings for the target image and adversarial prompt. Second, we iteratively optimize the image embedding using a Siamese semantic network guided by prompt-aligned cues (e.g., "luxury," "premium quality"), with multiplicative fusion modulated by a spatial layout mask [Chaitanya et al., 2020, Lee et al., 2021, Li et al., 2022b]. Third, we apply perceptual and semantic losses, including LPIPS Zhang et al. [2018], to preserve identity and realism during optimization. Fourth, the modified embedding is decoded into a final image using Stable Diffusion. This process yields images that are visually plausible yet semantically manipulated to influence downstream agent decisions.

We rigorously evaluate TRAP using automated multimodal LLM-based evaluators performing N-way comparisons, benchmarking against standard baselines such as: diffusion generation without optimization, Simultaneous Perturbation Stochastic Approximation (SPSA) [Spall, 1987], and Bandit [Ilyas et al., 2018b]. Our evaluation spans six leading multimodal models: LLaVA-1.5-34B [Liu et al., 2023a], Gemma3-8B [Mesnard et al., 2025], Mistral-small-3.1-24B and Mistral-small-3.2-24B, GPT-4o [OpenAI et al., 2024], and CogVLM [Wang et al., 2024a], covering both open- and closed-source architectures. A robust agent should maintain its original objectives and resist such preference manipulations, as allowing adversarial inputs to fundamentally alter decision-making processes undermines the agent's intended purpose and trustworthiness. Our results demonstrate that TRAP significantly outperforms these baselines in shifting autonomous agents' preferences toward adversarially manipulated images. These findings provide strong evidence of critical vulnerabilities within autonomous multimodal systems, raising important safety and trustworthiness concerns about relying exclusively on autonomous perception without adequate safeguards.

Ultimately, this paper serves as both a critical demonstration of potential security vulnerabilities in agentic AI systems and a call to action for developing more robust multimodal alignment, perception safeguards, and adversarial defenses within autonomous systems deployed in real-world environments.

## 2 Related Works

### 2.1 Adversarial Attacks on Agentic Systems

Recent advancements in autonomous, agentic AI systems have revealed critical vulnerabilities to adversarial manipulations. Yang et al. [2024] and Wang et al. [2024b] demonstrate backdoor-injection attacks that subtly corrupt agent behavior, misleading web agents during decision making. Similarly, Wu et al. [2024] and Liao et al. [2025] reveal that carefully crafted prompt injections can lead agents to take unintended actions, ranging from disclosing private information to leaking web contents. However, these approaches require extensive access to either the environment's internals or the model's parameters, an assumption rarely met in practical settings. In contrast, our work addresses a more realistic attack setting in which the adversary can only manipulate their own indexed input elements (e.g., images or prompts) without any knowledge of the underlying environment code or model weights.

### 2.2 Image-based Adversarial Attacks

Image-based adversarial attacks have been extensively studied, focusing primarily on neural network classifiers. White-box attacks, which require full gradient access, range from the foundational Fast Gradient Sign Method (FGSM) [Goodfellow et al., 2015] to iterative optimization methods. These include Projected Gradient Descent (PGD) [Madry et al., 2019] and more advanced approaches like Carlini & Wagner (C&W) [Carlini and Wagner, 2016] and DeepFool [Moosavi-Dezfooli et al., 2016a], which formulate the attack as a constrained optimization problem to find minimal perturbations.

In the more constrained black-box setting, query-based methods estimate gradients using techniques like finite differences (SPSA [Uesato et al., 2018], ZOO [Chen et al., 2017b]) or evolutionary strategies (Bandits [Ilyas et al., 2018b], NES [Ilyas et al., 2018a]). Other approaches bypass gradient estimation entirely, such as the query-efficient Square Attack [Andriushchenko et al., 2020] or decision-based methods like Boundary Attack [Brendel et al., 2018]. Different attack modalities have also been explored, including localized Adversarial Patches [Brown et al., 2017] for physical-world robustness and Universal Adversarial Perturbations (UAP) [Moosavi-Dezfooli et al., 2017] that are image-agnostic.

These approaches typically aim to induce misclassification or alter model output minimally and undetectably. Our work expands upon these methodologies by leveraging semantic manipulation through text-guided diffusion models, aiming to influence model decisions at a deeper semantic level.

### 2.3 Diffusion Models and Semantic Image Manipulation

Diffusion-based generative models, such as Stable Diffusion, have emerged as powerful tools for high-fidelity image synthesis guided by textual prompts. These models encode rich semantic relationships between text and image domains, enabling precise manipulation of generated images. For example, Wang et al. [2023], Dai et al. [2024] fine-tune latent diffusion codes to introduce targeted changes in object appearance, such as color shifts or texture edits, that mislead classification models while remaining imperceptible to humans. Liu et al. [2023b] guides the reverse diffusion process using free-text instructions, producing adversarial examples that adhere to natural language descriptions and allow fine-grained control over semantic attributes. Zhai et al. [2023a] shows that poisoning only a small subset of text–image pairs during training can backdoor large text-to-image diffusion models at the pixel, object, or style level, embedding hidden triggers that activate under specific prompts. In contrast to these methods, our approach uses only the model embeddings to generate adversarial images, without requiring access to model parameters or training data for the diffusion model.

## 3 Preliminaries

### 3.1 Stable Diffusion and Textual Guidance

Modern diffusion models, such as Stable Diffusion [Rombach et al., 2022b], operate not in the high-dimensional pixel space $\mathbb{R}^{C \times H \times W}$ but in a computationally cheaper, perceptually-equivalent latent space. This is achieved using a Variational Autoencoder (VAE), which consists of an encoder $\mathcal{E}_{\text{VAE}}$ and a decoder $\mathcal{D}_{\text{VAE}}$. The encoder compresses a full-resolution image $x_0$ into a smaller latent

representation $z_0 = \mathcal{E}_{\text{VAE}}(x_0)$. The decoder is trained to reconstruct the image from this latent, $x_0 \approx \mathcal{D}_{\text{VAE}}(z_0)$.

The core diffusion process, which learns to reverse a noising process, is trained entirely within this latent space. The generative model is a denoising network $\epsilon_\theta$ that is trained to predict the noise $\epsilon$ that was added to a noisy latent $z_t$ at timestep $t$. This denoising network can be conditioned on external information, such as a text prompt. To achieve this, the text prompt $p$ is first converted into a $d$-dimensional embedding $c = E_T(p)$ using a text encoder. This conditioning vector $c$ is then fed into the $\epsilon_\theta$ network, typically via cross-attention layers, allowing the text to guide the denoising process: $\epsilon_\theta(z_t, t, c)$.

To strengthen this guidance, modern samplers use Classifier-Free Guidance (CFG) [Ho and Salimans, 2022]. At each denoising step $t$, the network makes two predictions: one conditioned on the prompt's embedding $c$, $\epsilon_\theta(z_t, t, c)$, and one unconditioned, $\epsilon_\theta(z_t, t, \emptyset)$, which uses a null-text embedding $\emptyset$. The model then extrapolates in the "direction" of the prompt by mixing these two predictions where the scalar $w$ is the guidance scale (CFG scale) and a higher $w$ forces the generation to adhere more strictly to the text prompt $c$:

$$\hat{\epsilon}_t = \epsilon_\theta(z_t, t, \emptyset) + w \cdot \big(\epsilon_\theta(z_t, t, c) - \epsilon_\theta(z_t, t, \emptyset)\big)$$

### 3.2 CLIP

Contrastive Language–Image Pre-training (CLIP) [Radford et al., 2021] learns to connect images and text by projecting them into a single, shared embedding space. It achieves this by training two encoders simultaneously: a vision encoder $E_V$ and a text encoder $E_T$. These encoders map an image $x$ and a text prompt $p$ into a shared $d$-dimensional space, producing normalized embeddings $v = E_V(x)$ and $u = E_T(p)$, where $d$ is the dimensionality of this shared space.

The training objective is contrastive: given a batch of $(x, p)$ pairs, the model is trained to maximize the similarity for correctly matched pairs while minimizing the similarity for all incorrect unmatched pairs. This similarity is measured by the scaled dot product of their embeddings, $v_i^\top u_j / \tau$.

$$\mathcal{L}_{\text{CLIP}} = -\frac{1}{2}\left[\log \frac{\exp(v^\top u / \tau)}{\sum_j \exp(v^\top u_j / \tau)} + \log \frac{\exp(v^\top u / \tau)}{\sum_i \exp(v_i^\top u / \tau)}\right],$$

where $\tau$ is a learned temperature parameter. This symmetric function is the average of two cross-entropy losses. The first term (image-to-text loss) treats the task as classifying the correct text embedding $u$ (the "positive" sample) for a given image embedding $v$, out of all $N$ text embeddings $\{u_j\}$ (also including the "negative" samples) in the batch. The second term (text-to-image loss) does the opposite, classifying the correct $v$ for a given $u$ from all $N$ image embeddings $\{v_i\}$.

Minimizing $\mathcal{L}_{\text{CLIP}}$ forces the model to maximize the softmax probability for the correct pair in both directions. This simultaneously maximizes the dot product of matched pairs (the numerators) and minimizes the dot products of all unmatched, "negative" pairs (the denominators). This dynamic "pulls" semantically related image-text embeddings together and "pushes" unrelated pairs apart, creating a shared space where proximity equates to semantic similarity.

### 3.3 Learned Perceptual Similarity

The Learned Perceptual Image Patch Similarity (LPIPS) metric [Zhang et al., 2018] compares deep features extracted from a pre-trained classification network $\mathcal{F}$, which is used as a fixed feature extractor.

Given two images, a reference $x_r$ and a perturbed image $x_p$, both are passed through the network $\mathcal{F}$. LPIPS extracts the activation maps $h_r^l, h_p^l \in \mathbb{R}^{C_l \times H_l \times W_l}$ from a set of $L$ different layers. These activations are then normalized along the channel dimension denoted $\hat{h}^l$. The squared $\ell_2$ distance is computed between these normalized activations, and this distance is then averaged across all spatial locations. The final LPIPS score $d(x_r, x_p)$ is a weighted sum of these layer-wise distances:

$$d(x_r, x_p) = \sum_{l=1}^{L} w_l \cdot \frac{1}{H_l W_l} \sum_{h,w} \big\| \hat{h}_r^l(h, w) - \hat{h}_p^l(h, w) \big\|_2^2$$

The small weights $w_l$ are themselves learned to best correlate with human perceptual judgments. A lower LPIPS score $d$ signifies a higher perceptual similarity between the two images.

# 4 Methods

## 4.1 Problem Formulation

Modern autonomous agentic AI systems increasingly rely on multimodal models that integrate vision and language to make decisions with minimal human oversight. These systems are deployed in real-world applications such as e-commerce, navigation agents, and booking platforms, where the selected image directly triggers downstream actions [Davydova et al., 2025, Wang et al., 2024c], such as clicks, follow-up queries, or further reasoning steps, making this vision-language selection layer a key target for influencing agentic behavior [Li et al., 2025, Zhu et al., 2024].

Formally, we consider an agent driven by a multimodal model $M$. Consistent with standard contrastive retrieval architectures [Radford et al., 2021], we assume $M$ contains a text encoder $E_T$ and a vision encoder $E_V$. The agent receives a text prompt $p$ and a set of $n$ candidate images $\{x_i\}_{i=1}^n$. The model computes an embedding for the prompt $e_{\text{text}} = E_T(p)$, and for each image $e_{\text{image}}(x_i) = E_V(x_i)$. The model selects the image $x_i$ that maximizes an internal relevance score of $f_M$, defined as their cosine similarity:

$$M\big(p, \{x_i\}_{i=1}^n\big) = \arg \max_{i \in \{1,\dots,n\}} f_M(p, x_i), \qquad f_M(p, x_i) = \cos\big(e_{\text{text}}, e_{\text{image}}(x_i)\big)$$

We consider an attacker whose objective is to force the agent to select a specific target image. The attacker controls a single index $t$, i.e., a target image $x_{\text{target}} = x_t$, and can replace it with an adversarially modified version $x_{\text{adv}}$. The remaining $n-1$ images, which form the competitor set $\{x_{\text{comp}}^{(i)}\}_{i=1}^{n-1} = \{x_i\}_{i \neq t}$, cannot be modified by the attacker. The attack goal is to produce $x_{\text{adv}}$ such that the agent selects it. Equivalently, we seek to increase the selection probability, i.e.:

$$M(p, \{x_{\text{adv}}\} \cup \{x_{\text{comp}}^{(i)}\}_{i=1}^{n-1}) = x_{\text{adv}}, \qquad \Pr\big[f_M(p, x_{\text{adv}}) > \max_{x \in \{x_{\text{comp}}^{(i)}\}_{i=1}^{n-1}} f_M(p, x)\big]. \quad (1)$$

This optimization is subject to the constraint that $x_{\text{adv}}$ remains perceptually similar to the original image $x_{\text{target}}$. Formally, we require $d(x_{\text{adv}}, x_{\text{target}}) \leq \epsilon$, where $d$ is a perceptual distance metric (e.g., LPIPS) and $\epsilon$ is a small perceptual budget. This constraint is necessary for the attack to be viable in a real-world system. An unconstrained attack, where an attacker simply generates a new image $x_{\text{adv}}$ to perfectly match the prompt $p$, would be trivially defeated. Such an image would fail basic platform-level integrity checks, such as visual hash-based deduplication or anomaly detection [Hao et al., 2021, Wu et al., 2023, Zhou et al., 2018], which would flag the large semantic gap between the original and modified content as a fraudulent replacement, not a subtle perturbation. The perceptual constraint $d(x_{\text{adv}}, x_{\text{target}}) \leq \epsilon$ is explicitly designed to bypass these defenses by ensuring $x_{\text{adv}}$ is perceived as a benign modification of $x_{\text{target}}$.

This attack is performed under a realistic black-box threat model as the attacker has no access to the model's weights, parameters, or gradients. Their only capability is to query the model $M$ and observe the final selection, mimicking an attacker who can only interact with the agent's public application. By systematically probing the vision-language selection layer of the agentic systems, we expose and characterize vulnerabilities in multimodal agentic systems that could be exploited to threaten the reliability and fairness of downstream decision-making.

## 4.2 TRAP Framework

To expose the susceptibility of multimodal agents to this threat model, we propose TRAP, a novel black-box optimization framework that modifies only the target image to induce consistent selection by AI agents. TRAP diverges from traditional pixel-level perturbations by operating in CLIP's latent space rather than directly modifying image pixels. This allows us to steer high-level semantics in a model-agnostic manner using a surrogate representation aligned with vision-language reasoning. This choice is motivated by the increasing robustness of modern systems to low-level noise and the limitations of existing pixel-based attacks in black-box, semantic decision settings [Yang et al., 2022, Li et al., 2023, Goodfellow et al., 2015, Madry et al., 2019].

TRAP takes as input a target image $x_{\text{target}}$, competitor images $\{x_{\text{comp}}^{(i)}\}_{i=1}^{n-1}$, and an attacker-chosen guidance prompt, $p_{\text{pos}}$, which describes the high-level concepts to be injected. It is critical to distinguish $p_{\text{pos}}$ from the unknown, user-provided prompt $p$ that the agent receives at inference time. The threat model does not assume knowledge of the exact $p$. Instead, the attacker chooses a $p_{\text{pos}}$ to act as a strong semantic proxy for a family of desirable user queries (e.g., using $p_{\text{pos}} =$ "steel-toe reinforced" to capture searches for both "durable boot" and "safe boot"). TRAP uses this $p_{\text{pos}}$ as the optimization target for its semantic alignment loss ($\mathcal{L}_{\text{sem}}$), taking advantage of the shared embedding space to ensure that optimizing for $p_{\text{pos}}$ generalizes to increase the selection probability for semantically similar $p$ queries. We pre-compute the embedding for this prompt as $e_{\text{pos}} = E_T(p_{\text{pos}})$.

The framework iteratively optimizes the image latent embedding $e_{\text{adv}}$, initialized from $E_V(x_{\text{target}})$, to maximize its alignment with the guidance prompt's embedding, $e_{\text{pos}}$. This approach is effective because most modern multimodal models (e.g., CLIP, ALIGN, BLIP) rank images by their similarity to the prompt in the shared embedding space; a higher alignment directly translates to a higher likelihood of selection.

While the precise architecture of the agent may be unknown, prior work demonstrates that adversarial examples crafted in CLIP space are transferable to other vision-language models due to shared embedding geometries and training objectives [Huang et al., 2025]. We therefore optimize $e_{\text{adv}}$ using a composite objective, and the final optimized embedding is decoded into the adversarial image $x_{\text{adv}}$. The following subsections detail each component of TRAP, and the complete process is summarized in Algorithm 1.

### 4.3 Guided Embedding Optimization in CLIP Space

Our goal is to find an optimal adversarial embedding $e_{\text{adv}}^*$ by minimizing a composite objective. The optimization is as such:

$$e_{\text{adv}}^* = \arg\min_{e_{\text{adv}}} \mathcal{L}_{\text{total}}(e_{\text{adv}}) \tag{2}$$

The total loss $\mathcal{L}_{\text{total}}$ depends on $e_{\text{adv}}$ as well as several other pre-computed constants. Since $e_{\text{adv}}$ is the only variable, we optimize this function via gradient descent. The objective is a weighted sum of three components where $\lambda_1, \lambda_2, \lambda_3$ are scalar hyperparameters:

$$\mathcal{L}_{\text{total}}(\cdot) = \lambda_1 \mathcal{L}_{\text{sem}}(\cdot) + \lambda_2 \mathcal{L}_{\text{dist}}(\cdot) + \lambda_3 \mathcal{L}_{\text{LPIPS}}(\cdot) \tag{3}$$

**Semantic Alignment Loss ($\mathcal{L}_{sem}$).** First, to influence the model's selection behavior in favor of $x_{adv}$, we leverage CLIP's joint image-text embedding space, where semantically related inputs are embedded nearby. By minimizing the cosine distance between $e_{adv}$ and the $\ell_2$-normalized positive prompt embedding $e_{pos}$, we inject high-level semantic meaning directly into the image representation:

$$\mathcal{L}_{\text{sem}}(e_{\text{adv}}, e_{\text{pos}}) = 1 - \cos(e_{\text{adv}}, e_{\text{pos}})$$

**Distinctive Feature Preservation Loss ($\mathcal{L}_{dist}$).** Minimizing $\mathcal{L}_{\text{sem}}$ alone could cause $e_{\text{adv}}$ to lose the image's unique identity. To prevent this, we introduce a Siamese semantic network $S_{\text{dist}}$. This network (e.g., a two-branch MLP) is designed to decompose a given embedding $e \in \mathbb{R}^d$ into two components: a common embedding $e_{\text{com}} \in \mathbb{R}^{d_c}$ and a distinctive embedding $e_{\text{dist}} \in \mathbb{R}^{d_d}$, where $d_c$ and $d_d$ are the dimensionalities of these two output feature spaces set within the network configurations as hyperparameters.

$$S_{\text{dist}}(e) = (e_{\text{com}}, e_{\text{dist}})$$

This loss penalizes the $\ell_2$ distance between the distinctive component of $e_{\text{adv}}$ and the distinctive component of the original $e_{\text{target}}$.

$$\mathcal{L}_{\text{dist}}(e_{\text{adv}}, e_{\text{target}}^{(\text{dist})}) = \|S_{\text{dist}}(e_{\text{adv}})[1] - e_{\text{target}}^{(\text{dist})}\|_2^2$$

Here, $S_{\text{adv}}(e_{\text{target}})[1]$ denotes the distinctive component of $S_{\text{dist}}$ (the second output of $S_{\text{dist}}$) applied to the current $e_{\text{adv}}$. The term $e_{\text{target}}^{(\text{dist})}$ is a pre-computed constant, $e_{\text{target}}^{(\text{dist})} = S_{\text{dist}}(E_V(x_{\text{target}}))[1]$. This constraint ensures that adversarial edits preserve identity-relevant features not captured by semantic alignment alone. Without it, optimization may overfit to prompt content, collapsing diverse inputs into visually indistinct representations (e.g., all "apple" images becoming generic red blobs). As demonstrated in prior work on multimodal attacks [Zhang et al., 2024a, Chen et al., 2025], targeting both shared and distinctive features increases adversarial effectiveness and transferability. Our loss

thus anchors the adversarial embedding to its unique identity while still allowing semantic guidance from the prompt.

This loss creates the implicit supervision for the decomposition. By penalizing any change in the distinctive branch, $\mathcal{L}_{\text{dist}}$ effectively "anchors" $e_{\text{dist}}$ to its original value. This forces the optimizer to channel the gradients from the other two losses ($\mathcal{L}_{\text{sem}}$ and $\mathcal{L}_{\text{LPIPS}}$) almost exclusively through the common branch $S_{\text{dist}}(e_{\text{adv}})[0]$. This push-pull dynamic is what isolates the prompt-driven semantic changes to $e_{\text{com}}$, while $e_{\text{dist}}$ preserves the image's unique identity.

**Perceptual Similarity Loss ($\mathcal{L}_{LPIPS}$).** Finally, to ensure the decoded image remains visually plausible, we apply a loss in pixel space. This requires a differentiable process to decode $e_{\text{adv}}$ into a candidate image $x_{\text{cand}}$ at each optimization step. This decoding pipeline itself has two parts.

First, we generate a semantic layout mask $A \in \mathbb{R}^{H \times W}$ using a lightweight MLP encoder-decoder $L$ to identify regions of interest:

$$A = L_{\text{dec}}\big(L_{\text{enc}}([e_{\text{pos}}, E_V(x_{\text{target}})])\big) \tag{4}$$

This initial mask $A$ is a "soft" heatmap indicating semantic relevance to the prompt, but it may be spatially imprecise. To improve localization and ensure edits are restricted to the primary subject, we refine $A$ using a pre-computed binary foreground mask, $F_{\text{seg}}$, obtained from a DeepLabv3 segmentation model [Chen et al., 2017a]. The final mask used for modulation is the element-wise product:

$$A_{\text{final}} = A \odot F_{\text{seg}} \tag{5}$$

Second, at each optimization step, we extract the common component from our trainable embedding, $e_{\text{com}} = S_{\text{dist}}(e_{\text{adv}})[0]$, and modulate it with the fixed, refined mask $A_{\text{final}}$ to get $e_{\text{mod}} = e_{\text{com}} \odot A_{\text{final}}$. This $e_{\text{mod}}$ is passed to a differentiable image decoder $SD$ (e.g., the VAE decoder from Stable Diffusion), conditioned on $p_{\text{pos}}$, to produce the candidate image $x_{\text{cand}} = SD(e_{\text{mod}}, p_{\text{pos}})$.

The LPIPS loss is the perceptual distance between this decoded candidate $x_{\text{cand}}$ and the original, unmodified target image $x_{\text{target}}$:

$$\mathcal{L}_{\text{LPIPS}}(x_{\text{cand}}, x_{\text{target}}) = \text{LPIPS}(x_{\text{cand}}, x_{\text{target}})$$

By tracing the dependencies, $x_{\text{cand}}$ is a function of $e_{\text{adv}}$, $A_{\text{final}}$, and $p_{\text{pos}}$. Since $A_{\text{final}}$ and $p_{\text{pos}}$ are pre-computed constants, $\mathcal{L}_{\text{LPIPS}}$ is an implicit function of $e_{\text{adv}}$. This creates a fully differentiable path from the pixel-space comparison back to our latent variable $e_{\text{adv}}$.

**Overall framework** The full optimization and final decoding process is summarized as:

$$
\begin{aligned}
e_{\text{adv}}^* &= \arg\min_{e_{\text{adv}}} \big[\lambda_1 \mathcal{L}_{\text{sem}} + \lambda_2 \mathcal{L}_{\text{dist}} + \lambda_3 \mathcal{L}_{\text{LPIPS}}\big], \\
A &= L_{\text{dec}}(L_{\text{enc}}([e_{\text{pos}}, E_V(x_{\text{target}})])), \qquad A_{\text{final}} = A \odot F_{\text{seg}}, \\
e_{\text{com}} &= S_{\text{dist}}(e_{\text{adv}})[0], x_{\text{adv}} = SD(e_{\text{com}} \odot A_{\text{final}}, p_{\text{pos}}).
\end{aligned}
\tag{6}
$$

with each step guided by semantic alignment, visual coherence, and layout-informed embeddings. Algorithm 1 in the Appendix summarizes the optimization process for generating an adversarial image given a target, a prompt, and a black-box agent model.

## 5 Experimental Methodology

### 5.1 Experimental Protocol

We evaluate our attack on 100 image-caption pairs from the popular COCO Captions dataset [Chen et al., 2015], simulating a black-box $n$-way selection setting. For each instance, a "bad image" is generated using a negative prompt created via Llama-3-8B [Grattafiori et al., 2024]. This image is verified to have an initial selection probability below the majority threshold when compared against $n-1$ competitors, ensuring a challenging starting point for optimization.

Adversarial optimization then runs for up to $K = 20$ outer iterations, each containing $T = 20$ inner gradient-based steps, stopping early if the success condition is met. To evaluate the final optimized image $x_{\text{adv}}$, we conduct $R = 100$ randomized trials. In each trial, $x_{\text{adv}}$ and the $n-1$ competitors are randomly ordered to mitigate positional bias [Tian et al., 2025], horizontally concatenated into

$I_{concat}$, and queried to the agent model $M$. The selection probability $P(x_{\text{adv}})$ is the fraction of these $R$ trials where $x_{\text{adv}}$ was chosen:

$$P(x_{\text{adv}}) = \frac{1}{R} \sum_{r=1}^{R} \mathbf{1}[M(I_{concat}) = x_{\text{adv}}]$$

The overall Attack Success Rate (ASR) is the percentage of the 100 COCO instances where this optimized image successfully crosses the majority threshold, i.e., $P(x_{\text{adv}}) > 1/n$.

## 5.2 Model and Implementation Details

All experiments are implemented in PyTorch. We use CLIP ViT-B/32 [Radford et al., 2021] for embedding extraction, with adversarial image decoding performed by Stable Diffusion v2.1 (base) through the Img2Img interface. The optimized image embedding is repeated across 77 tokens and injected as prompt embeddings into the UNet decoder.

The Siamese Semantic Network consists of two branches, each with two linear layers (512→1024), BatchNorm, and ReLU, trained to decompose CLIP image embeddings into common and distinctive features. The Layout Generator receives concatenated image and text embeddings (1536 dimensions), processes them via an encoder (linear layers: 1536→512→1024, ReLU), reshapes to (256, 2, 2), then upsamples with five transposed convolutional layers with ReLU and a final Sigmoid to generate a spatial mask $A \in \mathbb{R}^{H \times W}$, which is refined with DeepLabv3 segmentation to emphasize foreground. Optimization is performed with Adam (learning rate 0.005, 20 steps per iteration). Grid search is conducted over diffusion strength $[0.3, 0.8]$ and CFG $[2.0, 12.0]$ with initial values of 0.5 and 7.5, respectively. All experiments were run on a server with four NVIDIA A100-PCIE-40GB GPUs and a 48-core Intel Xeon Silver 4214R CPU. Average per-iteration optimization is around 520 seconds compared to around 376 seconds for SPSA and 110 seconds for Bandit.

# 6 Experimental Results

## 6.1 Main Findings

TRAP achieves the highest attack success rate (ASR) across all evaluated models, LLaVA-1.5 [Liu et al., 2023a], Gemma3 [Mesnard et al., 2025], Mistral-small [Mistral AI, 2025], GPT-4o [OpenAI et al., 2024], and CogVLM [Wang et al., 2024a]. As shown in Table 1, TRAP universally succeeds even from a challenging low-preference baseline (0–21% ASR), while traditional baselines like SPSA [Spall, 1987] and Bandit [Ilyas et al., 2018b] are ineffective (max 36% ASR). We also compare against recent embedding-space attacks: SSA_CWA [Chen et al., 2024], which evaluates robustness using embedding-based attacks, and SA_AET [Jia et al., 2025], which generates adversarial images by projecting text embeddings onto image embeddings. While these embedding-based methods perform better than traditional approaches, TRAP significantly outperforms all baselines. The attack's transferability is a key finding. It demonstrates high efficacy not only on expected contrastive models but also on CogVLM, which uses a non-contrastive architecture. Additionally, the attack transfers with high success to GPT-4o, a completely closed-source, black-box proprietary model.

Table 1: Comparison of adversarial attack effectiveness across evaluated methods and models.

| Method | LLaVA 1.5-34B | Gemma 3-8B | Mistral-small-3.1-24B | Mistral-small-3.2-24B | GPT-4o | CogVLM |
|---|---|---|---|---|---|---|
| Initial "bad image" | 21% | 17% | 14% | 6% | 0% | 8% |
| SPSA | 36% | 27% | 22% | 11% | 1% | 18% |
| Bandit | 6% | 2% | 1% | 0% | 0% | 0% |
| Stable Diffusion | 24% | 18% | 18% | 7% | 0% | 20% |
| SSA_CWA | 65% | 42% | 28% | 18% | 8% | 4% |
| SA_AET | 85% | 67% | 61% | 55% | 12% | 42% |
| **TRAP** | **100%** | **100%** | **100%** | **99%** | **63%** | **94%** |

We further assess TRAP against standard defenses and an adversarially trained LLaVA variant (Robust-LLaVA). As summarized in Table 2, TRAP remains highly effective. Applying addi-

tive Gaussian noise to the generated adversarial images has a negligible effect on ASR (TRAP + Gaussian Noise). Caption-level filters applied post-attack, such as CIDER [Xu et al., 2024] and MirrorCheck [Fares et al., 2024], reduce ASR but do not eliminate the attack's success, particularly on the robust model. Figure 3 provides qualitative examples of successful attacks generated by TRAP.

Table 2: Robustness of TRAP Attack Under Various Defense Mechanisms and Adversarial Training.

| COCO | LLaVA-1.5-34B | Gemma3-8B | Mistral-small-3.1-24B | Mistral-small-3.2-24B | Robust-LLaVA |
|---|---|---|---|---|---|
| TRAP | 100% | 100% | 100% | 97% | 92% |
| TRAP + Gaussian Noise | 100% | 100% | 100% | 96% | 92% |
| TRAP + CIDER | 100% | 100% | 96% | 90% | 85% |
| TRAP + MirrorCheck | 100% | 98% | 88% | 82% | 74% |

## 6.2 Robustness to System Prompt and Temperature

We further evaluate robustness to system-prompt phrasing and sampling temperature. Table 3 reports $\Delta$ASR when using four different rephrased system prompt variants compared to the baseline ASR achieved with the standard system prompt used in our main experiments, with shifts confined to low single digits and averages near zero, indicating strong generalization under rewording. Overall, as long as the instruction semantics are preserved, TRAP remains stable to superficial template changes. Additional ablation study can be found in Appendix A.

Table 3: Impact of system prompt variations on attack success rate (ASR). $\Delta$ASR is the average deviation from baseline.

| Model | Variant 1 | Variant 2 | Variant 3 | Variant 4 | Avg. $\Delta$ASR |
|---|---|---|---|---|---|
| LLaVA-1.5-34B | +2% | −1% | +4% | +1% | +2% |
| Gemma3-8B | −2% | +1% | −3% | −1% | −1% |
| Mistral-small-3.1-24B | +1% | +2% | −1% | +0% | +1% |

Figure 2 plots the ASR against the required margin over the majority choice threshold ($1/n$), comparing performance at two distinct decoding temperatures: $T = 0.1$ (left plot, representing near-deterministic output) and $T = 0.7$ (right plot, representing more stochastic output). Across both temperature settings, the optimized TRAP attack (solid lines) consistently maintains a high ASR, significantly outperforming the unoptimized baseline (dashed lines) across all evaluated models. The ASR curves for the optimized attack show minimal variation between the low-temperature ($T = 0.1$) and high-temperature ($T = 0.7$) settings, confirming that the attack's effectiveness is robust to changes in the agent's sampling temperature and ensuring reliability under both deterministic and stochastic generation conditions.

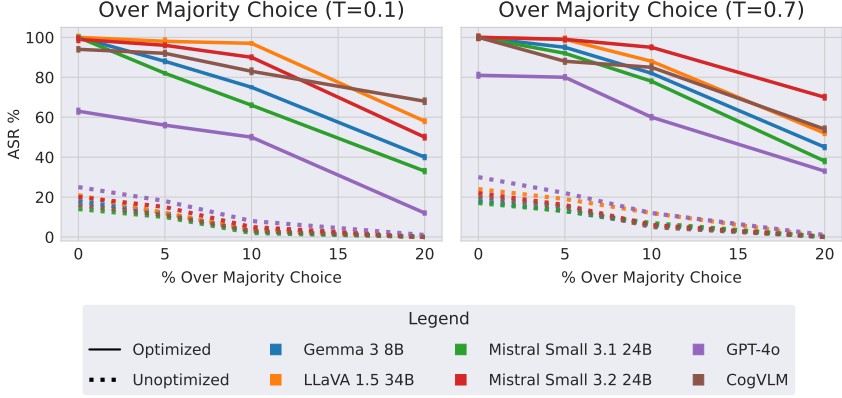

Figure 2: Attack success rate under different sampling temperatures.

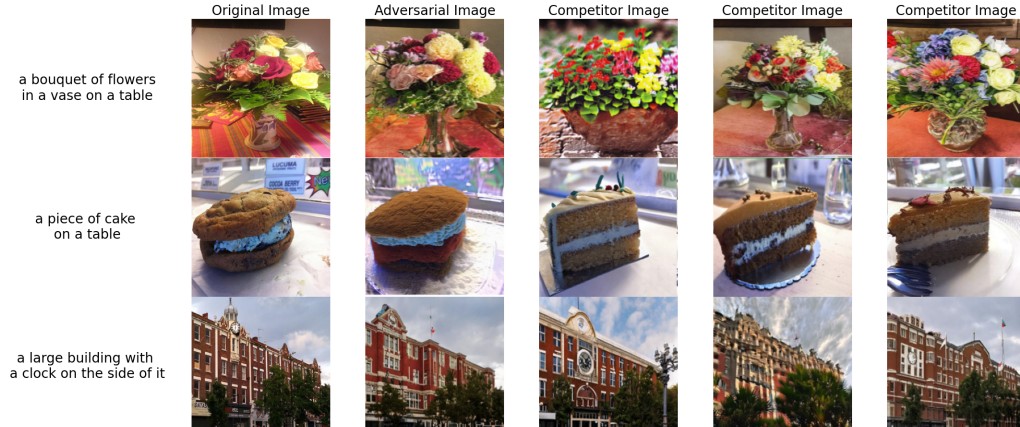

Figure 3: Qualitative examples of successful attacks. Each row shows a user-facing scenario where the attacker modifies a target image (left) into an adversarial variant (second column), evaluated against three unmodified competitors (right).

# 7 Discussion

We expose a critical vulnerability in agentic AI: visually subtle, semantically guided attacks reliably mislead VLMs even under black-box constraints. TRAP consistently induces decision-level preference redirection across all evaluated MLLMs, far exceeding traditional pixel-based and standard diffusion baselines. This expands the attack surface in real-world multimodal agents.

**Key takeaways**: (1) semantic attacks are effective and transferable; (2) semantic attacks are robust to prompt and sampling noise and remain visually plausible; (3) the vulnerability generalizes across MLLMs; (4) existing defenses overlook this threat class.

However, the broader significance lies in what such attacks enable. By adversarially altering an image to match a high-level semantic concept, attackers could manipulate agentic behavior in downstream tasks, causing selection of malicious UI elements, misleading product recommendations, hijacked retrieval in chat agents, or sabotage of autonomous perception pipelines. More importantly, these edits remain visually natural and can be deployed in black-box settings, making them difficult to detect or attribute compared to previous methods. This work challenges the assumption that robustness can be measured solely through pixel-space perturbations, emphasizing the need for embedding-level defenses and semantic-level robustness criteria.

# 8 Limitations

While TRAP demonstrates strong performance, several limitations must be acknowledged. We assume that the agent relies on contrastive vision-language similarity, an assumption supported by current architectures but potentially less valid in future systems that move completely away from contrastive reasoning or incorporate stronger semantic defenses than the one tested. The success of our method also depends on the quality of auxiliary components such as the layout mask and diffusion model; performance may degrade on edge cases or under constrained resources. Finally, TRAP is more computationally intensive than pixel-level attacks, due to its reliance on iterative optimization and generative decoding. While this cost can be amortized in offline scenarios or reduced via model distillation, scalability to real-time applications remains an open challenge.

## Acknowledgements

We thank the anonymous reviewers for their thoughtful and constructive feedback. This work was supported by funding through NSF Grants No. CCF-2238079, CCF-2316233, CNS-2148583 anda Research Gift from Amazon AGI Labs.

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

# A Appendix

## A.1 Algorithm

---

**Algorithm 1** TRAP Framework

---

**Require:** Target image $x_{\text{target}}$, guidance prompt $p_{\text{pos}}$, competitor set $\{x_{\text{comp}}^{(i)}\}_{i=1}^{n-1}$, black-box agent $M$

**Require:** Encoders $E_V, E_T$; Decoder $SD$; Siamese Network $S_{\text{dist}}, L$; Seg. model $F_{\text{model}}$

**Require:** Hyperparameters $\lambda_1, \lambda_2, \lambda_3$; Loops $K, T$; Evals $R$

**Ensure:** Optimized adversarial image $x_{\text{adv}}$

1: Initialize $best\_score \leftarrow 0$, $n \leftarrow |\{x_{\text{comp}}^{(i)}\}_{i=1}^{n-1}| + 1$
2: $e_{\text{target}} \leftarrow E_V(x_{\text{target}})$
3: $e_{\text{pos}} \leftarrow E_T(p_{\text{pos}})$
4: $e_{\text{dist}}^{(\text{target})} \leftarrow S_{\text{dist}}(e_{\text{target}})[1]$
5: $A \leftarrow L(e_{\text{pos}}, e_{\text{target}})$
6: $F_{\text{seg}} \leftarrow F_{\text{model}}(x_{\text{target}})$
7: $A_{\text{final}} \leftarrow A \odot F_{\text{seg}}$
8: $x_{\text{adv}} \leftarrow x_{\text{target}}$
9: **for** $k = 1$ to $K$ **do**
10:     $e_{\text{adv}} \leftarrow e_{\text{target}}$
11:     **for** $t = 1$ to $T$ **do**
12:         $(e_{\text{com}}, e_{\text{dist\_adv}}) \leftarrow S_{\text{dist}}(e_{\text{adv}})$
13:         $e_{\text{mod}} \leftarrow e_{\text{com}} \odot A_{\text{final}}$
14:         $x_{\text{cand}} \leftarrow SD(e_{\text{mod}}, p_{\text{pos}})$
15:         $\mathcal{L}_{\text{sem}} \leftarrow 1 - \cos(e_{\text{adv}}, e_{\text{pos}})$
16:         $\mathcal{L}_{\text{dist}} \leftarrow \|e_{\text{dist\_adv}} - e_{\text{dist}}^{(\text{target})}\|_2^2$
17:         $\mathcal{L}_{\text{LPIPS}} \leftarrow \text{LPIPS}(x_{\text{cand}}, x_{\text{target}})$
18:         $\mathcal{L} \leftarrow \lambda_1 \mathcal{L}_{\text{sem}} + \lambda_2 \mathcal{L}_{\text{dist}} + \lambda_3 \mathcal{L}_{\text{LPIPS}}$
19:         Update $e_{\text{adv}}$ using gradient descent on $\mathcal{L}$
20:     **end for**
21:     $(e_{\text{com\_final}}, \_) \leftarrow S_{\text{dist}}(e_{\text{adv}})$
22:     $x_{\text{final\_cand}} \leftarrow SD(e_{\text{com\_final}} \odot A_{\text{final}}, p_{\text{pos}})$
23:     $P_{\text{select}} \leftarrow \text{EstimateProb}(M, p_{\text{pos}}, x_{\text{final\_cand}}, \{x_{\text{comp}}^{(i)}\}_{i=1}^{n-1}, R)$
24:     **if** $P_{\text{select}} > best\_score$ **then**
25:         $best\_score \leftarrow P_{\text{select}}$
26:         $x_{\text{adv}} \leftarrow x_{\text{final\_cand}}$
27:     **end if**
28:     **if** $best\_score \geq 1/n$ **then**
29:         **break**
30:     **end if**
31: **end for**
32: **return** $x_{adv}$

---

## A.2 Ablation Study

### A.2.1 Iterative Refinement

To provide a more intuitive understanding of our method's behavior, we present a series of qualitative examples in Figure 4. This figure visualizes the output of our iterative refinement process on three distinct image-prompt pairs, showcasing the progressive transformation of the input images over a sequence of iterations.

Figure 4 illustrates the model's ability to apply semantically-guided changes that evolve in complexity as the number of iterations increases. Each row corresponds to a specific prompt: "a piece of cake on a table" , "a large building with a clock on the side of it" , and "a bouquet of flowers in a vase on a table".

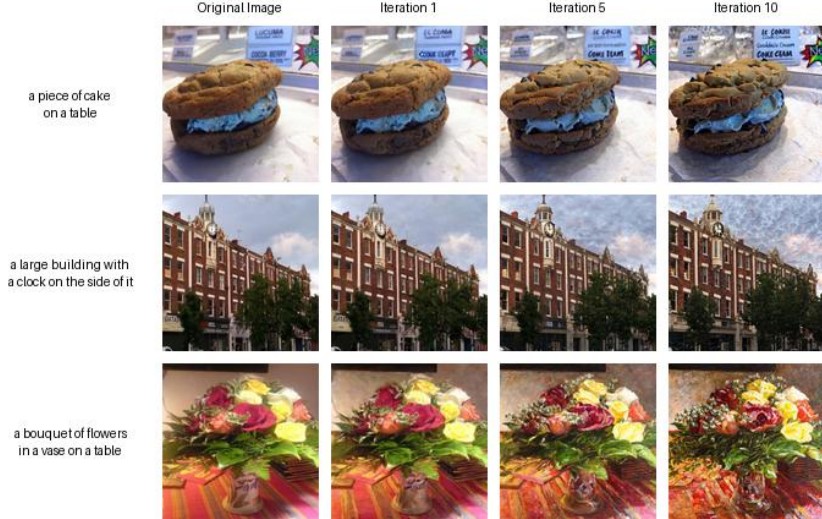

Figure 4: Qualitative results of the iterative image generation process. Each row begins with an original source image and a corresponding text prompt, followed by the generated outputs at iterations 1, 5, and 10. The examples demonstrate a range of transformations, from subtle refinement to significant stylistic alteration.

### A.2.2 Evaluation on Additional Datasets

We evaluate TRAP on two additional distributions, Flickr8k_sketch and ArtCapt [Lu et al., 2024]. Across both settings and across diverse VLM backbones (LLaVA-1.5-34B, Gemma3-8B, Mistral variants, GPT-4o), TRAP consistently outperforms baseline attacks (Tables 4 and 5). This pattern suggests the objective is not tied to a single model family and transfers across architectures and data regimes.

Table 4: TRAP Attack Success Across Sketch and Abstract Image Dataset (Flickr8k_sketch).

| Flickr8k_sketch | LLaVA-1.5-34B | Gemma3-8B | Mistral-small 3.1-24B | Mistral-small 3.2-24B | GPT-4o |
|---|---|---|---|---|---|
| SPSA | 41% | 33% | 31% | 26% | 16% |
| Bandit | 4% | 3% | 1% | 0% | 0% |
| Stable Diffusion (no opt.) | 20% | 22% | 18% | 11% | 4% |
| **TRAP** | **100%** | **100%** | **100%** | **96%** | **72%** |

Table 5: TRAP Attack Success Rates Across Artistic Image Styles (ArtCap Dataset).

| ArtCap | LLaVA-1.5-34B | Gemma3-8B | Mistral-small 3.1-24B | Mistral-small 3.2-24B | GPT-4o |
|---|---|---|---|---|---|
| SPSA | 33% | 29% | 20% | 21% | 18% |
| Bandit | 7% | 3% | 0% | 0% | 0% |
| Stable Diffusion (no opt.) | 25% | 20% | 17% | 10% | 2% |
| **TRAP** | **100%** | **100%** | **100%** | **95%** | **58%** |

### A.2.3 Variation in Hyperparameters

We vary the relative weights of the perceptual, semantic, and distinctive objectives. Increasing the perceptual and semantic terms preserves strong performance, whereas aggressively scaling the distinctive term can reduce transfer on some targets (Table 6). These trends align with the intuition

that over-emphasizing uniqueness may hurt cross-model alignment. On the other hand, decreasing the perceptual and distinctive terms maintains strong performance, while removing the semantic term showed a significant decrease in the performance (Table 7).

Table 6: Effect of Increasing Lambda Coefficients on Attack Success Rate.

| | LLaVA-1.5-34B | Gemma3-8B | Mistral-small-3.1-24B | Mistral-small-3.2-24B |
|---|---|---|---|---|
| **Perceptual Loss** $1.0 \rightarrow 1.5$ | 100% | 100% | 100% | 98% |
| **Semantic Loss** $0.5 \rightarrow 1.0$ | 100% | 100% | 94% | 92% |
| **Distinctive Loss** $0.3 \rightarrow 0.8$ | 88% | 70% | 72% | 65% |

Table 7: Effect of Decreasing Lambda Coefficients on Attack Success Rate.

| | LLaVA-1.5-34B | Gemma3-8B | Mistral-small-3.1-24B | Mistral-small-3.2-24B |
|---|---|---|---|---|
| **Perceptual Loss** $1.0 \rightarrow 0.8$ | 100% | 100% | 98% | 94% |
| **Semantic Loss** $0.5 \rightarrow 0.0$ | 90% | 82% | 77% | 70% |
| **Distinctive Loss** $0.3 \rightarrow 0.0$ | 100% | 100% | 91% | 88% |

### A.2.4 Embedding Model Choice

Substituting a range of image embedding backbones (from ViT-B/32 to larger SigLIP-style models [Zhai et al., 2023b] and Jina-CLIP [Koukounas et al., 2024]) yields essentially the same outcome (Table 8), indicating that TRAP is not unduly sensitive to the representation family. This stability simplifies deployment since the attack does not hinge on a particular feature extractor.

Table 8: TRAP Attack Success Rates Ablation on Different Embedding Models.

| Embedding model | LLaVA-1.5-34B | Gemma3-8B | Mistral-small-3.1-24B |
|---|---|---|---|
| ViT-B/32 | 100% | 100% | 100% |
| timm/ViT-SO400M-14-SigLIP-384 | 100% | 100% | 99% |
| jinaai/jina-clip-v2 | 100% | 100% | 97% |

### A.2.5 Different Diffusion Model

We also swap the image generator among popular Stable Diffusion variants. Performance remains consistent across SD-2.1 [Rombach et al., 2022a], SD-XL [Podell et al., 2023], and SD-1.5 (Table 9), suggesting the synthesis backend is not a critical factor for the attack.

### A.2.6 E-commerce Webpage Scenario

In a more realistic page-level setting, TRAP maintains a sizable margin over prior strategies (Table 10). While absolute rates are lower than in curated benchmarks, the relative gains persist under stronger content controls.

Table 9: Attack Success Rates Ablation on Different Stable Diffusion Image Generators.

| Generator | LLaVA-1.5-34B | Gemma3-8B | Mistral-small-3.1-24B |
|---|---|---|---|
| stable-diffusion-2-1 | 100% | 100% | 100% |
| stable-diffusion-xl-base-1.0 | 100% | 100% | 100% |
| stable-diffusion-v1-5 | 100% | 100% | 100% |

Table 10: Attack Success Rates of TRAP and Baselines in E-commerce Webpage Scenarios.

| COCO | LLaVA-1.5-34B | Gemma3-8B | Mistral-small-3.1-24B | Mistral-small-3.2-24B |
|---|---|---|---|---|
| SPSA | 10% | 7% | 1% | 0% |
| Bandit | 0% | 0% | 0% | 0% |
| Stable Diffusion | 13% | 0% | 3% | 0% |
| SSA_CWA | 20% | 17% | 13% | 12% |
| SA_AET | 30% | **24%** | 7% | 3% |
| **TRAP** | **51%** | **50%** | **27%** | **17%** |

### A.2.7 Efficiency Considerations

Finally, we report end-to-end compute. As expected for optimization-based attacks, TRAP is more expensive per sample than lightweight heuristics (Table 11). This overhead can be mitigated with standard engineering (e.g., early stopping, caching, adaptive step sizes) and batching.

Table 11: Total Computation Time per Sample.

| | Computational Time per Step (s) | # of Steps per Iteration | # of Iterations | Total Computational Time per Sample (s) |
|---|---|---|---|---|
| SPSA | 0.94 | 20 | 20 | 376s |
| Bandit | 0.00022 | 10,000 | 50 | 110s |
| Stable Diffusion | 4.6 | 1 | 1 | 4.6s |
| TRAP | 1.3 | 20 | 20 | 520s |

