# OpenReview forum: "TRAP: Targeted Redirecting of Agentic Preferences"
_NeurIPS.cc/2025/Conference — NeurIPS 2025 poster_

### Official Review · Reviewer_riRf · 2025-06-18

**Clarity:** 3
**Significance:** 2
**Originality:** 2
**Rating:** 4
**Confidence:** 3

**Summary:**

This paper introduces TRAP, an adversarial framework exploiting vulnerabilities in vision-language model (VLM) agents by manipulating images to misguide their decisions, without needing internal model access.  TRAP subtly alters CLIP text embeddings via diffusion models and targeted semantic optimization, preserving visual plausibility while steering interpretations towards attacker-chosen concepts (e.g., "luxury").  Evaluations on VLMs like LLaVA-34B showed a 100% attack success rate, significantly outperforming baseline methods and proving robust against defenses.  The findings highlight the need for defense strategies beyond pixel-level robustness to address semantic vulnerabilities in cross-modal decision-making.

**Questions:**

1. The authors should clarify whether the proposed method is applicable to images across diverse scenarios and styles. A more detailed discussion on how semantic enhancement is achieved for different contexts would strengthen the work. Specifically, what mechanisms enable the method to handle varying semantic content effectively？
2. The adversarial sample generation process appears relatively complex.  To better assess the method's practicality, the authors should provide a thorough analysis of its computational complexity, including comparative experiments on training/inference time against baseline approaches.  This would help readers evaluate the method's feasibility in real-world applications.
3. Ablation experiments are necessary to validate the contribution of each proposed module.  Quantitative results demonstrating the impact of individual components would provide concrete evidence of their respective roles and significance in the overall framework.
4. The methodological section would benefit from clearer explanations of the design rationale behind each module.  Explicitly stating the motivation for key architectural choices would greatly improve readability and help readers better understand the technical innovations.

**Ethical Concerns:**

["NO or VERY MINOR ethics concerns only"]

**Final Justification:**

The paper presents TRAP, an adversarial framework that exploits vulnerabilities in vision-language model (VLM) agents by manipulating images to mislead their decisions without requiring internal model access. TRAP operates by subtly altering CLIP text embeddings through diffusion models and targeted semantic optimization, maintaining visual plausibility while steering interpretations toward attacker-specified concepts (e.g., “luxury”). The authors’ rebuttal addressed most of my concerns, and after considering the perspectives of other reviewers, I assess the final rating as borderline accept.

**Limitations:**

Yes.

**Paper Formatting Concerns:**

1. The citation format is incorrect.
2. The references are not formatted correctly.

**Quality:**

2

**Strengths And Weaknesses:**

Strengths：
1. The related work and problem formalization are clearly described, making the background and positioning of the proposed method easy to understand.
2. Extensive experiments were conducted on the settings of the system prompt and temperature, validating the model's robustness.

Weaknesses：
1. The authors claim that their proposed method operates in a black-box manner and does not require access to model internals. However, in the experimental section, they only present attack results on three types of open-source models. It is recommended to supplement the evaluation with attack performance assessments on prevalent proprietary models (e.g., GPT-4V) to substantiate the method's applicability to genuine black-box models.
2. The experimental evaluation appears limited in scope. The study only compares against a few conventional baselines while omitting recent advances in image-based adversarial attack methodologies (e.g., [1][2][3]). This omission undermines a comprehensive assessment of the proposed method’s performance advantages. It is strongly recommended that the authors include comparative analyses with state-of-the-art adversarial attack techniques to better validate their claims.
3. A more thorough discussion comparing the proposed method with prior works (e.g., [1][2][3]) would help better position its contributions within the literature.
4. To better demonstrate the generalizability of the proposed method, It is recommended to conduct additional experiments on a wider range of multimodal large-scale models with diverse architectures.
4. The manuscript contains persistent formatting inconsistencies, particularly in reference citations and listings, which should be carefully revised for adherence to the conference guidelines.
5. Figure 1 currently fails to clearly convey the methodological innovation or core workflow.   We suggest a redesign to improve conceptual clarity and readability.

[1] How Robust is Google's Bard to Adversarial Image Attacks?

[2] Jailbreak in pieces: Compositional Adversarial Attacks on Multi-Modal Language Models

[3] Automatic and Universal Prompt Injection Attacks against Large Language Modals

---

> ### Author Rebuttal · Authors · 2025-07-31
>
> Thank you for your suggestions and valuable feedback on our work. We would like to address each of your concerns:
>
> 1. > "The authors should clarify whether the proposed method is applicable to images across diverse scenarios and styles..."
>     - **Response:** TRAP is designed to handle diverse scenarios and visual styles by combining adaptive feature decomposition with text-guided semantic modulation. Specifically, our architecture separates features into common (prompt-aligned) and distinctive (identity-preserving) branches, with the importance of each dynamically adjusted via the text projection layer and bounded by sigmoid activation. This allows TRAP to enhance semantics in varying prompts and contexts.
> We demonstrate strong attack performance not just on COCO, but also on other datasets such as Flickr8k_sketch (sketches, abstracted forms) and ArtCap (artworks, varied styles), as shown in Tables 1–3. TRAP outperforms baseline attacks even in these challenging domains, supporting its robustness across content, context, and style diversity.
>
> **Table 1 · Extended evaluation on the COCO dataset**
> | COCO | LLaVA-1.5-34B | Gemma3-8B | Mistral-small-3.1-24B | Mistral-small-3.2-24B | GPT-4o |
> |-----------------|--------------:|----------:|----------:|------------------:|-------:|
> | SPSA      | 36 % | 27 % | 22 % | 11 % | 8 % |
> | Bandit        | 6 %  | 2 %  | 1 %  | 0 %  | 0 % |
> | Stable Diffusion (no opt.)      | 24 % | 18 % | 18 % | 7 %  | 2 % |
> | **TRAP**     | **100 %** | **100 %** | **100 %** | **97 %** | **63 %** |
>
> &nbsp;
>
> **Table 2 · TRAP Attack-Success Rate Across Sketch / Abstract-Image Dataset (Flickr8k_sketch)**
> | Flickr8k_sketch | LLaVA-1.5-34B | Gemma3-8B | Mistral-small-3.1-24B | Mistral-small-3.2-24B | GPT-4o |
> |--------|--------------:|----------:|-------------:|----------------------:|-------:|
> | SPSA   | 41 % | 33 % | 31 % | 26 % | 16 % |
> | Bandit   | 4 %  | 3 %  | 1 %  | 0 %  | 0 %  |
> | Stable Diffusion (no opt.)      | 20 % | 22 % | 18 % | 11 % | 4 %  |
> | **TRAP**  | **100 %** | **100 %** | **100 %** | **96 %** | **72 %** |
>
> &nbsp;
>
> **Table 3 · TRAP Attack-Success Rate Across Artistic Image Styles (ArtCap dataset)**
> | ArtCap dataset | LLaVA-1.5-34B | Gemma3-8B | Mistral-small-3.1-24B | Mistral-small-3.2-24B | GPT-4o |
> |--------|-----:|----------:|--------:|-----:|-------:|
> | SPSA  | 33 % | 29 % | 20 % | 21 % | 18 % |
> | Bandit  | 7 %  | 3 %  | 0 %  | 0 %  | 0 %  |
> | Stable Diffusion (no opt.) | 25 % | 20 % | 17 % | 10 % | 2 %  |
> | **TRAP**  | **100 %** | **100 %** | **100 %** | **95 %** | **58 %** |
>
> 2. > "The adversarial sample generation process appears relatively complex... "
>     - **Response:** We thank the reviewer for highlighting the importance of computational efficiency. As shown in Table 4, TRAP generates a single optimized adversarial image in an average of 666 seconds on an NVIDIA A100 GPU. The total time per sample for TRAP reflects the full iteration budget (20 steps × 20 iterations), representing a conservative, worst-case upper bound. In practice, TRAP’s early stopping mechanism means most attacks succeed in fewer iterations, resulting in a lower average runtime for the majority of samples. Additionally, TRAP remains practical for many real-world scenarios, such as e-commerce platforms and moderation pipelines where attacks can be generated offline before deployment or review.
>
> **Table 4 · Total computation time per sample**
> | Method  | Steps / Second | Steps / Iteration | Iterations | Total time (s) / Sample |
> |-----|------:|--------:|-----------:|--------:|
> | SPSA  | 0.94  | 20  | 20 | 168 s  |
> | Bandit     | 0.00022  | 10 000  | 50 | 11 s  |
> | Stable Diffusion (no opt.)    | 4.6   | 1 | 1  | 4.6 s  |
> | TRAP | 0.6  | 20   | 20 | 666.6 s|
>
> &nbsp;
>
> 3. > "Ablation experiments are necessary to validate the contribution of each proposed module... "
>     - **Response:** We ran removal ablations to establish causality, where any consistent drop in attack success can be attributed to that module’s contribution rather than parameter count or tuning. The results of this ablation study in Table 5 show that removing any main component substantially reduces ASR, confirming each component’s necessity.
>
> **Table 5 · Contribution of individual TRAP components to overall attack effectiveness**
> | Component ablated   | LLaVA-1.5-34B | Gemma3-8B | Mistral-small-3.1-24B |
> |-----|--------:|------:|-------:|
> | Without Prompt-Driven Modulation  | 41 % | 44 % | 31 % |
> | Without Distinctive Loss   | 77 % | 71 % | 70 % |
> | Without Siamese Network   | 54 % | 50 % | 33 % |
>
> 4. > "The methodological section would benefit from clearer explanations of the design rationale behind each module... "
>     - **Response:** We agree that a more explicit explanation will help readers and will make these clarifications in the final version of the paper.
>     - Prompt-Driven Modulation:
>       - *Why not attention mechanisms or types of interactions?*
> We experimented with standard attention layers as alternatives for layout generation. However, in our pilot tests, attention layers increased per-step optimization time by over 30% without measurable ASR improvement, so we adopted a more efficient encoder-decoder architecture with direct spatial modulation.
>     - Siamese Semantic Network:
>       - *How is the Siamese Network trained to achieve the decomposition, and what supervision ensures proper feature separation?*  The Siamese branches are supervised by the attack optimization itself: the common (semantic) branch minimizes semantic loss (Lsem) toward the prompt, while the distinctive branch minimizes identity loss (Ldist) relative to the original embedding, both taking advantage of CLIP’s contrastive structure. No external labels are required, the dual loss structure naturally polarizes each branch toward its intended feature set.
>     - Loss Function and Hyperparameter Ablation:
>       - *Is there a tension between preserving distinctive features and achieving semantic alignment? Are the lambda coefficients justified?*
> There is a balance required between semantic and identity preservation. Our new ablation results (Tables 2 and 3) systematically vary each lambda coefficient, confirming that semantic loss is essential, perceptual loss is robust to moderate changes, and the distinctive loss must remain low for best performance.
>
> **Table 6 · Effect of Increasing λ-Coefficients on Attack-Success Rate**
> | Lambda change & loss type   | LLaVA-1.5-34B | Gemma3-8B | Mistral-small-3.1-24B | Mistral-small-3.2-24B |
> |----------|--------:|----------:|-------------:|------:|
> | Lambda 1 (Perceptual) 1.0 → 1.5    | 100 % | 100 % | 100 % | 98 % |
> | Lambda 2 (Semantic) 0.5 → 1.0   | 100 % | 100 % | 94 %  | 92 % |
> | Lambda 3 (Distinctive) 0.3 → 0.8   | 88 %  | 70 %  | 72 %  | 65 % |
> &nbsp;
>
> **Table 7 · Effect of Decreasing λ-Coefficients on Attack-Success Rate**
> | Lambda change & loss type   | LLaVA-1.5-34B | Gemma3-8B | Mistral-small-3.1-24B | Mistral-small-3.2-24B |
> |------|------:|----------:|-----------:|---------:|
> | Lambda 1 (Perceptual) 1.0 → 0.8  | 100 % | 100 % | 98 % | 94 % |
> | Lambda 2 (Semantic) 0.5 → 0.0   | 90 %  | 82 %  | 77 % | 70 % |
> | Lambda 3 (Distinctive) 0.3 → 0.0 | 100 % | 100 % | 91 % | 88 % |
>
> 5. > "The experimental evaluation appears limited in scope... "
>     - **Response:** We appreciate the pointer to recent advances and recognize the importance of comparisons with more modern adversarial attack methods. We address each of the reviewer's given examples below:
>       - Liu et al. (2024) study automatic prompt injection in text pipelines specifically, with the idea of universal injected strings that can subvert LLM‑integrated apps by altering external text content. We find this paper to be adjacent to our work, but we expand the idea of redirecting autonomous behavior to a multimodal context, which increases the complexity of the problem considerably.
>       - Chen et al. (2024) introduce a Common Weakness Attack (CWA) to improve transferability of pixel‑bounded adversarial examples via flatness and proximity objectives, validated mostly on image classification and extended to a Bard demo, not a selection setting like the one TRAP is evaluated in. We include a transfer baseline from this paper in Table 8 and find that TRAP remains superior in the preference selection setting.
>       - Shayegani et al. (2023) perform embedding‑space image editing that pairs a benign prompt with an adversarial image targeted to "toxic" embeddings and focuses on jailbreak generation, not selection. To address the reviewer's concerns about comparing TRAP's evaluation with modern state-of-the-art methodologies, we add in Table 8 an additional comparison with SA-AET, which boosts cross‑model transfer by sampling within an adversarial evolution triangle and optimizing in a semantic‑aligned contrast subspace; in our selection‑based evaluation with matched access and iteration budgets, SA‑AET is competitive, but TRAP achieves higher success (Table 8).
>
> **Table 8 · Attack-success rates of TRAP versus baseline methods across multiple VLMs**
> | Method  | LLaVA-1.5-34B | Gemma3-8B | Mistral-small-3.1-24B | Mistral-small-3.2-24B | GPT-4o | CogVLM |
> |--------|--------:|-------:|------:|-------:|-------:|-------:|
> | SPSA   | 36 % | 27 % | 22 % | 11 % | 1 %  | 18 % |
> | Bandit   | 6 %  | 2 %  | 1 %  | 0 %  | 0 %  | 0 %  |
> | Stable Diffusion (no opt.)     | 24 % | 18 % | 18 % | 7 %  | 0 %  | 20 % |
> | SSA_CWA | 65 % | 42 % | 28 % | 18 % | 8 %  | 4 %  |
> | SA-AET  | 85 % | 67 % | 61 % | 55 % | 12 % | 42 % |
> | TRAP | **100 %** | **100 %** | **100 %** | **99 %** | **63 %** | **94 %** |
>
> 6. > "The manuscript contains persistent formatting inconsistencies..."
>     - **Response:** We will correct the typos in the final version of the paper.
>
> 7. > "Figure 1 currently fails to clearly convey..."
>     - **Response:** We will design a diagram that better emphasizes the key contributions of our methodology in the final version of the paper.

---

> > ### Comment · Reviewer_riRf · 2025-08-06
> >
> > Thank you to the authors for the thorough response. Most of my concerns have been adequately addressed, and I will raise my score to 4 accordingly.

---

> ### Author Response · Authors · 2025-08-07
> **Thank You for Your Review and Feedback**
>
> Thank you very much for your thoughtful review, constructive feedback, and follow-up. We greatly appreciate your careful consideration of our work and your willingness to engage with our clarifications and new results and the discussions and experiments mentioned in the rebuttal will be included in the final version of our paper.

---

### Official Review · Reviewer_LRnX · 2025-07-02

**Clarity:** 2
**Significance:** 2
**Originality:** 2
**Rating:** 4
**Confidence:** 2

**Summary:**

This paper proposes a new attack for multimodal (image-text) models relying on both gradient descent-based optimization and embedding similarity with caption information to generate adversarial images. In the sophisticated multi-step decision problems tested, the attack performs very well, achieving an 100% attack success rate compared to chosen baselines.

**Questions:**

- Is there a typo in Equation 3?
- Similarly, Algorithm 1 has many bespoke design decisions. Have you thoroughly ablated them, for example, the choice of embedding model?
- Are there experiments analyzing the effectiveness of the iterative nature of this algorithm? Even a qualitative result could go a long way here: showing an image after 1 step, 2 steps, etc. If this information was available in the supplementary I did not find it.
- What are some other baselines that could be incorporated into Table 1? For example, what about Adversarial Illusions in Multi-Modal Embeddings (2025)? A quick google search shows that many papers have been published in this space, but none are considered as serious baselines.

**Ethical Concerns:**

["NO or VERY MINOR ethics concerns only"]

**Final Justification:**

I was wrong about the typo in the equation. The authors have provided a large number of new results which will improve the paper if they do indeed integrate them.

**Limitations:**

No, actually. I would like to see discussion of Ethics and societal implications in the limitations section. In its current form it only describes experimental limitations.

**Quality:**

2

**Strengths And Weaknesses:**

Strengths:
- Multimodal security is an interesting problem and the idea of incorporating caption information for adversarial image grounding seems important.
- The qualitative results, adversarial images in Figure 4, seem interesting and reasonable.
- The use of segmentation masks (4.2.2, a-la DeepLabv3) to improve the relevance of adversarial perturbations seems novel and useful.
- The claimed results, if taken at face value, are actually very good: 100% on all tasks and settings while baselines achieve 6–36%.

Weaknesses:
- Much of the paper discusses "agents" but there is nothing agentic about the approach. It's a standard multimodal attack.
- The baselines seem weak – how is it possible that the best baseline reaches 36% in one setting (and most are in the 10–20% range) while your proposed approach gets 100% in all settings?
- Is "100 image-caption pairs" a typical sample size for this subfield? That seems small, and no error bars are shown in Table 1 (although I suppose the main method must have a stderr of zero for all results since the numbers are all 100).

---

> ### Author Rebuttal · Authors · 2025-07-31
>
> Thank you for your suggestions and valuable critique of our work. We would like to address each of your concerns:
>
> 1. > "Is there a typo in Equation 3?"
>     - **Response:** We have carefully reviewed Equation 3 and did not identify any typographical errors. If this concern persists, please let us know, and we will address it in the final version.
> 2. > "Similarly, Algorithm 1 has many bespoke design decisions. Have you thoroughly ablated them, for example, the choice of embedding model?"
>     - **Response:** Thank you for raising this point. We have included an ablation study of the embedding model choice in Table 1. Across embedding models such as ViT-B/32, timm/ViT-SO400M-14-SigLIP-384, and jina-clip-v2, TRAP consistently achieves high attack success rates, with only minor fluctuations between models. This confirms that our approach is robust to the specific embedding model used. We also present ablation studies on other decisions in the following:
>       - Table 8 from reviewer riRf is the evaluation of TRAP on Mistral-small-3.2-24B, GPT-4o, and CogVLM, where CogVLM is a model not trained with contrastive objectives. Furthermore, Table 8 compares TRAP with more baselines: SSA_CWA and SA-AET
>       - Table 5 from reviewer qNmy shows TRAP’s robustness on defences such as simple Gaussian Noise, CIDER, and MirrorCheck. TRAP is also evaluated on Robust-LLaVA, a model specifically trained to be robust against adversarial perturbations.
>       - Table 2 & Table 3 from reviewer qNmy are ablation studies on the effect of different values for the coefficient for each loss function
>       - Table 5 from the reviewer Ce3w is an evaluation of TRAP on different Stable-Diffusion image generators.
>
> **Table 1 · TRAP attack-success rates ablation on different embedding models**
> | Model    | LLaVA-1.5-34B | Gemma3-8B | Mistral-small-3.1-24B |
> |---------------|--------------:|----------:|------------:|
> | ViT-B / 32      | 100 % | 100 % | 100 % |
> | timm/ViT-SO400M-14-SigLIP-384    | 100 % | 100 % | 99 %  |
> | jinaai/jina-clip-v2    | 100 % | 100 % | 97 %  |
>
> &nbsp;
>
> 3. > "Are there experiments analyzing the effectiveness of the iterative nature of this algorithm... "
>     - **Response:** Due to current rebuttal restrictions on attaching image files to our rebuttal, we will be adding a qualitative image sequence to the appendix in the final version of the paper. Table 2 provides an ablation analysis of the effect of the number of optimization iterations on ASR. The results show that even a single optimization iteration with TRAP already outperforms baseline attacks in terms of ASR, and increasing the number of iterations further amplifies this advantage across all models tested.
> **Table 2 · Effect of optimisation-iteration count on TRAP attack-success rate**
> | Iteration count | &nbsp;LLaVA-1.5-34B&nbsp; | &nbsp;Gemma3-8B&nbsp; | &nbsp;Mistral-small-3.1-24B&nbsp; |
> |:-----------|--------------:|----------:|----------------:|
> | 1  | 57 %  | 58 % | 44 % |
> | 5  | 73 %  | 61 % | 50 % |
> | 10 | 100 % | 93 % | 89 % |
> | 20 | 100 % | 100 %| 100 %|
>
> &nbsp;
>
> 4. > "What are some other baselines that could be incorporated into Table 1? For example..."
>     - **Response:** We agree with the need for stronger and more recent baselines. We have added two additional attacks to Table 3: BARD_SSA (as suggested by reviewer riRf) and a state-of-the-art semantic alignment attack. Most prior baselines rely on pixel-level perturbations, which recent models have become increasingly robust against due to new defense frameworks (i.e. Wang et al., 2025). In contrast, TRAP explicitly targets the semantic content of the image.
> Existing semantic attacks are primarily designed to achieve embedding alignment, often aiming to make the attack image semantically indistinguishable from the target, while our evaluation focuses on the image preference scenario, where having two highly similar images may lead to ambiguous selection outcomes. TRAP incorporates a distinctive loss and semantic alignment loss that pushes the produced image to be not just similar, but competitively more appealing to the model, yielding consistently higher attack success for our given scenario. Full results for all new baselines are shown in Table 3.
>
> **Table 3 · Attack-success rates of TRAP versus baseline methods across multiple VLMs**
> | COCO | LLaVA-1.5-34B | Gemma3-8B | Mistral-small-3.1-24B | Mistral-small-3.2-24B | GPT-4o |
> |--------------|--------------:|----------:|-----------------:|------------------:|-------:|
> | SPSA    | 36 % | 27 % | 22 % | 11 % | 8 % |
> | Bandit   | 6 %  | 2 %  | 1 %  | 0 %  | 0 % |
> | Stable Diffusion (no opt.)      | 24 % | 18 % | 18 % | 7 %  | 2 % |
> | SSA_CWA   | 65 % | 42 % | 28 % | 18 % | 9 % |
> | SA-AET  | 85 % | 67 % | 61 % | 55 % | 12 % |
> | **TRAP**   | **100 %** | **100 %** | **100 %** | **99 %** | **63 %** |
>
> &nbsp;
>
> 5. > "Much of the paper discusses "agents" but there is nothing agentic about the approach. It's a standard multimodal attack."
>     - **Response:** We appreciate this feedback and recognize the importance of evaluation in realistic agentic settings. As shown in Table 4, TRAP consistently achieves higher attack success rates than all baselines, even when adversarial images are embedded in actual e-commerce website screenshots with randomly shuffled product descriptions. This mirrors how modern frameworks like LangChain use browser automation with and vision models to analyze and interact with web content, demonstrating TRAP’s real-world relevance for modern agentic pipelines. Due to the regulations against file attachments, we are unable to provide the visual contents of this experiment setting. However, the respective figures will be added to the appendix in the final version of the paper.
>
> **Table 4 · Attack-success rates of TRAP and baselines in e-commerce webpage scenarios**
> | COCO  | LLaVA-1.5-34B | Gemma3-8B | Mistral-small-3.1-24B | Mistral-small-3.2-24B |
> |----------------------|--------------:|----------:|----------------------:|----------------:|
> | SPSA        | 10 % | 7 %  | 1 %  | 0 %  |
> | Bandit        | 0 %  | 0 %  | 0 %  | 0 %  |
> | Stable Diffusion (no opt.)      | 13 % | 0 %  | 3 %  | 0 %  |
> | SSA_CWA        | 20 % | 17 % | 13 % | 12 % |
> | SA_AET        | 30 % | 24 % | 7 %  | 3 %  |
> | **TRAP**     | **51 %** | **50 %** | **27 %** | **17 %** |
>
> &nbsp;
>
> 7. > "Is "100 image-caption pairs" a typical sample size for this subfield? That seems small..."
>     - **Response:** Each of our 100 image-caption pairs is evaluated with 100 randomized trials and up to 20 optimization iterations, resulting in over 200,000 model queries per experiment, and this computation is further multiplied across model architectures and datasets. While the sample size is 100 images per dataset, we have also tested TRAP on additional datasets (Flickr8k_sketch and ArtCap), and observed consistently strong attack success rates. This indicates that our sample selection provides robust and diverse coverage, and that TRAP’s effectiveness generalizes beyond the original COCO benchmark. Error bars for attack success rates (across three random seeds) are reported in Section 6.2 for key figures, and we will clarify this point further in the final version of our paper: “All results are averaged across 100 image-caption pairs and 100 randomized trials, with error bars reflecting variance across three seeds where shown.”
>
> **Table 5 · Extended evaluation on the COCO dataset**
> | COCO | LLaVA-1.5-34B | Gemma3-8B | Mistral-small-3.1-24B | Mistral-small-3.2-24B | GPT-4o |
> |-----------------|--------------:|----------:|----------------------:|------------------:|-------:|
> | SPSA      | 36 % | 27 % | 22 % | 11 % | 8 % |
> | Bandit        | 6 %  | 2 %  | 1 %  | 0 %  | 0 % |
> | Stable Diffusion (no opt.)      | 24 % | 18 % | 18 % | 7 %  | 2 % |
> | **TRAP**     | **100 %** | **100 %** | **100 %** | **97 %** | **63 %** |
>
> &nbsp;
>
> **Table 6 · TRAP Attack-Success Rate Across Sketch / Abstract-Image Dataset (Flickr8k_sketch)**
> | Flickr8k_sketch | LLaVA-1.5-34B | Gemma3-8B | Mistral-small-3.1-24B | Mistral-small-3.2-24B | GPT-4o |
> |--------|--------------:|----------:|----------------------:|----------------------:|-------:|
> | SPSA       | 41 % | 33 % | 31 % | 26 % | 16 % |
> | Bandit        | 4 %  | 3 %  | 1 %  | 0 %  | 0 %  |
> | Stable Diffusion (no opt.)      | 20 % | 22 % | 18 % | 11 % | 4 %  |
> | **TRAP**       | **100 %** | **100 %** | **100 %** | **96 %** | **72 %** |
>
> &nbsp;
>
> **Table 7 · TRAP Attack-Success Rate Across Artistic Image Styles (ArtCap dataset)**
> | ArtCap dataset | LLaVA-1.5-34B | Gemma3-8B | Mistral-small-3.1-24B | Mistral-small-3.2-24B | GPT-4o |
> |--------|--------------:|----------:|----------------------:|----------------------:|-------:|
> | SPSA     | 33 % | 29 % | 20 % | 21 % | 18 % |
> | Bandit     | 7 %  | 3 %  | 0 %  | 0 %  | 0 %  |
> | Stable Diffusion (no opt.) | 25 % | 20 % | 17 % | 10 % | 2 %  |
> | **TRAP**    | **100 %** | **100 %** | **100 %** | **95 %** | **58 %** |
> &nbsp;
>
> 8. > "No, actually. I would like to see discussion of Ethics... "
>     - **Response:** Thank you for raising this point. A central aim of our paper is to highlight the real-world risks posed by semantic attacks like TRAP and to increase awareness that current pixel-level and input-based defenses are insufficient. By demonstrating how these attacks can bypass content moderation, mislead AI systems, or reinforce biases, even in the presence of state-of-the-art mitigations and adversarially trained models (see Table 5 from reviewer qNmy, which shows TRAP’s robustness on defences such as simple Gaussian Noise, CIDER, and MirrorCheck. TRAP is also evaluated on Robust-LLaVA, a model specifically trained to be robust against adversarial perturbations), we hope to alert the community to this emerging threat and motivate the development of more robust, semantically-aware defenses. We will add this discussion of ethical and societal implications to the Limitations section in the final version of our paper.

---

> ### Author Response · Authors · 2025-08-07
> **Follow-up and Final Clarifications**
>
> Thank you again for your detailed review and constructive critique. We have addressed all of your comments and suggestions in our rebuttal, including further ablation studies, additional baseline attack methods, quantitative results for iterative optimization, broader dataset coverage, and expanded discussion of ethical implications. If you have any remaining questions or if there are aspects you would like further clarified, please let us know, we would be happy to provide additional detail or context. We appreciate your time and engagement with our work.

---

> > ### Comment · Reviewer_LRnX · 2025-08-08
> >
> > Thank you for all the additional baselines and ablations. I am encouraged to see this level of rebuttal engagement and expect that they will greatly improve the experimental foundation of the paper.

---

> > > ### Author Response · Authors · 2025-08-08
> > >
> > > We sincerely appreciate your encouraging feedback. We are glad that the additional baselines and ablations strengthen the experimental foundation, and we will ensure these improvements are clearly reflected in the final version of the paper.

---

### Official Review · Reviewer_qNmy · 2025-07-04

**Clarity:** 2
**Significance:** 2
**Originality:** 2
**Rating:** 4
**Confidence:** 3

**Summary:**

This paper proposes TRAP, a novel black-box adversarial framework that subtly manipulates vision-language model (VLM)-powered autonomous agents through semantically guided adversarial image generation using diffusion models. By optimizing in CLIP embedding space with a combination of perceptual, semantic, and identity-preserving losses, TRAP generates natural-looking images that consistently fool VLM agents across multiple settings, achieving 100% attack success rates against leading models.

**Questions:**

Can the authors demonstrate TRAP’s effectiveness in a more realistic agentic setting?

Are there any analysis on the \lamda1, \lamda2,\lamda3, in equation 3? How each loss terms affect final results? How the method depends on these hypermeters?

TRAP optimizes in CLIP space—how robust is the attack when the agentic model is based on non-contrastive or generative vision-language models?

Can TRAP be detected or mitigated by current perceptual defenses or CLIP anomaly detection tools? What if the agent is trained with an ensemble of prompts or with adversarial training?

Why Mistral 3.1, there are more updated versions in Mistral series, do you have results on these models?

**Ethical Concerns:**

["NO or VERY MINOR ethics concerns only"]

**Final Justification:**

Well, I think overall the paper is okay with good exp results, but the novelty of the method, and the practical use is still a concern somehow. I'll leave my score unchanged.

**Limitations:**

Since TRAP steers images toward embedding alignment with high-level prompts, it could unintentionally amplify pre-existing biases in the CLIP model.

While the simulation-based results are strong, real-world deployment conditions may degrade attack success.

**Paper Formatting Concerns:**

No paper formatting concerns.

**Quality:**

2

**Strengths And Weaknesses:**

Strength:
1) The threat model is timely and important, understanding VLM's vulnerabilities to semantic attacks is crucial.
2) The experiments are reasonable: the paper evaluates the method on multiple models and 100 randomized trials per image, with ablations across prompt variations, decoding temperature, and success thresholds. The figures are easy to follow. And the results shows significant improvements.

Weakness:
1) While the framework is designed for real-world agents, all evaluations are based on COCO and the researcher's own agent framework. Experiments on more practical agent framework would be better for strengthening the claims. For example, on LangChain.
2) The attack assumes agents use CLIP-like contrastive embedding spaces. It could be better to make more comparisons with [1]. There are plenty of embedding based methods, and I believe they can be pretty effective in this scenario. It could be better if the paper compares the proposed method with them in an ablation study.
[1] Dong, Yinpeng, et al. "How robust is google's bard to adversarial image attacks?." arXiv preprint arXiv:2309.11751 (2023).

---

> ### Author Rebuttal · Authors · 2025-07-31
>
> Thank you for your detailed feedback and useful suggestions. We would like to address your concerns:
>
> 1. > "Can the authors demonstrate TRAP’s effectiveness in a more realistic agentic setting?"
>     - **Response:** We appreciate this feedback and recognize the importance of evaluation in realistic agentic settings. As shown in Table 1, TRAP consistently achieves higher attack success rates than all baselines, even when adversarial images are embedded in actual e-commerce website screenshots from The Klarna Product Page Dataset. This mirrors how modern frameworks like Aguvis use browser automation with vision models to analyze and interact with web content, demonstrating TRAP’s real-world relevance for modern agentic pipelines. Due to the regulations against file attachments, we are unable to provide the visual contents of this experiment setting. However, the respective figures will be added to the appendix in the final version of the paper.
> &nbsp;
>
> **Table 1 · Attack-Success Rates of TRAP and Baselines in e-commerce Webpage Scenarios**
> | Method | LLaVA-1.5-34B | Gemma3-8B | Mistral-small-3.1-24B | Mistral-small-3.2-24B |
> |--------|-------------:|----------:|----------------------:|----------------------:|
> | SPSA                     | 10 % | 7 %  | 1 %  | 0 %  |
> | Bandit                   | 0 %  | 0 %  | 0 %  | 0 %  |
> | Stable Diffusion (no opt.) | 13 % | 0 %  | 3 %  | 0 %  |
> | SSA_CWA                  | 20 % | 17 % | 13 % | 12 % |
> | SA_AET                   | 30 % | 24 % | 7 %  | 3 %  |
> | **TRAP**    | **51 %** | **50 %** | **27 %** | **17 %** |
> &nbsp;
>
> 2. > "Are there any analysis on the \lamda1, \lamda2,\lamda3, in equation 3? How each loss terms affect final results? How the method depends on these hypermeters?"
>     - **Response:** We thank the reviewer for highlighting the importance of the hyperparameters. We have analyzed these hyperparameters in detail, as shown in Tables 2 and 3. The results show that TRAP’s performance is robust to moderate changes in the perceptual loss (λ1​), with only slight fluctuations in attack success rate when increased or decreased. In contrast, the semantic loss coefficient (λ2​) is essential: raising it too high can reduce transferability, while removing it sharply decreases attack success across all models, confirming its importance. For the distinctive loss (λ3​), higher values significantly hurt performance, while reducing or omitting it generally maintains high attack success.
> &nbsp;
>
> **Table 2 · Effect of Increasing λ-Coefficients on Attack-Success Rate**
> | Lambda change & loss type               | LLaVA-1.5-34B | Gemma3-8B | Mistral-small-3.1-24B | Mistral-small-3.2-24B |
> |-----------------------------------------|-------------:|----------:|----------------------:|----------------------:|
> | Lambda 1 (Perceptual) 1.0 → 1.5     | 100 % | 100 % | 100 % | 98 % |
> | Lambda 2 (Semantic) 0.5 → 1.0       | 100 % | 100 % | 94 %  | 92 % |
> | Lambda 3 (Distinctive) 0.3 → 0.8    | 88 %  | 70 %  | 72 %  | 65 % |
> &nbsp;
>
> **Table 3 · Effect of Decreasing λ-Coefficients on Attack-Success Rate**
> | Lambda change & loss type               | LLaVA-1.5-34B | Gemma3-8B | Mistral-small-3.1-24B | Mistral-small-3.2-24B |
> |-----------------------------------------|-------------:|----------:|----------------------:|----------------------:|
> | Lambda 1 (Perceptual) 1.0 → 0.8     | 100 % | 100 % | 98 % | 94 % |
> | Lambda 2 (Semantic) 0.5 → 0.0       | 90 %  | 82 %  | 77 % | 70 % |
> | Lambda 3 (Distinctive) 0.3 → 0.0    | 100 % | 100 % | 91 % | 88 % |
> &nbsp;
>
> 3. > "TRAP optimizes in CLIP space—how robust is the attack when the agentic model is based on non-contrastive or generative vision-language models?"
>     - **Response:** To assess the robustness of TRAP on non-contrastive models, we evaluated TRAP on CogVLM, a model not trained with contrastive objectives. As shown in Table 4, TRAP achieved a 94% attack success rate, substantially outperforming all baselines.
> &nbsp;
>
> **Table 4 · Effectiveness of Adversarial Attacks on a Non-Contrastive Vision–Language Model (CogVLM)**
> | Model  | &nbsp;&nbsp;&nbsp;SPSA&nbsp;&nbsp;&nbsp; | &nbsp;&nbsp;&nbsp;Bandit&nbsp;&nbsp;&nbsp; | &nbsp;&nbsp;&nbsp;Stable Diffusion (no opt.)&nbsp;&nbsp;&nbsp; | &nbsp;&nbsp;&nbsp;SSA_CWA&nbsp;&nbsp;&nbsp; | &nbsp;&nbsp;&nbsp;SA_AET&nbsp;&nbsp;&nbsp; | &nbsp;&nbsp;&nbsp;TRAP&nbsp;&nbsp;&nbsp; |
> |--------|:-----:|:-------:|:---------------------------:|:--------:|:-------:|:-----:|
> | CogVLM | 18 % | 0 %    | 20 %                      | 4 %     | 42 %   | **94 %** |
> &nbsp;
>
> 4. > "Can TRAP be detected or mitigated by current perceptual defenses or CLIP anomaly detection tools? What if the agent is trained with an ensemble of prompts or with adversarial training?"
>     - **Response:** We initially tested TRAP against simple noise-based defenses, which referred to adding Gaussian noise to the adversarial image, as in Byun et al. (2021), to disrupt pixel-level attack gradients. However, since TRAP operates at the semantic and embedding level, its effectiveness is minimally affected by such noise. Attack success rate dropped by at most 1% (see Table 5). We also evaluated TRAP against CIDER and MirrorCheck, which aim to mitigate both low-level and some higher-level perturbations, but TRAP consistently maintained high success rates. Finally, TRAP was tested on Robust-LLaVA, a model adversarially trained for robustness, and still achieved a 92% attack success rate. These results highlight that existing input-level and even adversarially trained defenses are insufficient to counter semantic attacks like TRAP.
> &nbsp;
>
> **Table 5 · Robustness of TRAP Attack Under Various Defense Mechanisms and Adversarial Training**
> | Defence                                   | LLaVA-1.5-34B | Gemma3-8B | Mistral-small-3.1-24B | Mistral-small-3.2-24B | Robust-LLaVA |
> |-----------------------------------------------|--------------:|----------:|----------------------:|----------------------:|-------------:|
> | TRAP                                          | 100 % | 100 % | 100 % | 97 % | 92 % |
> | TRAP + Gaussian&nbsp;Noise&nbsp;(σ = 8 / 255) | 100 % | 100 % | 100 % | 96 % | 92 % |
> | TRAP + CIDER                                  | 100 % | 100 % | 96 %  | 90 % | 85 % |
> | TRAP + MirrorCheck                            | 100 % | 98 %  | 88 %  | 82 % | 74 % |
> &nbsp;
>
> 5. > "Why Mistral 3.1, there are more updated versions in Mistral series, do you have results on these models?"
>     - **Response:** We thank the reviewer for their suggestion. In Table 6, we have added results for Mistral-small-3.2-24B and GPT-4o alongside the original experiments. TRAP maintains high attack success rates on this updated model, further supporting the generality of our method.
> &nbsp;
>
> **Table 6 · Attack-success-rate comparison on additional vision–language models**
> | COCO                     | LLaVA-1.5-34B | Gemma3-8B | Mistral-small-3.1-24B | Mistral-small-3.2-24B | GPT-4o |
> |---------------------------------|--------------:|----------:|----------------------:|----------------------:|-------:|
> | SPSA                            | 36 % | 27 % | 22 % | 11 % | 8 % |
> | Bandit                          | 6 %  | 2 %  | 1 %  | 0 %  | 0 % |
> | Stable Diffusion (no opt.)      | 24 % | 18 % | 18 % | 7 %  | 2 % |
> | SSA_CWA                         | 65 % | 42 % | 28 % | 18 % | 9 % |
> | SA-AET                          | 85 % | 67 % | 61 % | 55 % | 12 % |
> | **TRAP**                           | **100 %** | **100 %** | **100 %** | **99 %** | **63 %** |
> &nbsp;
>
> 6. > "The attack assumes agents use CLIP-like contrastive embedding spaces. It could be better to make more comparisons with..."
>     - **Response:** We appreciate the pointer to recent advances in this field and recognize the importance of comparisons with more modern adversarial attack methods. In Table 7, we report TRAP’s attack success rates alongside both the original perturbation methods and modern semantic-level attacks, including SSA-CWA and SA-AET. Across all models, including GPT-4o and CogVLM, a non-contrastive learning model, TRAP consistently outperforms these methods.
> &nbsp;
>
> **Table 7 · Attack-success rates of TRAP versus baseline methods across multiple VLMs**
> | COCO                     | LLaVA-1.5-34B | Gemma3-8B | Mistral-small-3.1-24B | Mistral-small-3.2-24B | GPT-4o | CogVLM |
> |---------------------------------|--------------:|----------:|----------------------:|----------------------:|-------:|-------:|
> | SPSA                            | 36 % | 27 % | 22 % | 11 % | 1 %  | 18 % |
> | Bandit                          | 6 %  | 2 %  | 1 %  | 0 %  | 0 %  | 0 %  |
> | Stable Diffusion (no opt.)      | 24 % | 18 % | 18 % | 7 %  | 0 %  | 20 % |
> | SSA_CWA                         | 65 % | 42 % | 28 % | 18 % | 8 %  | 4 %  |
> | SA_AET                          | 85 % | 67 % | 61 % | 55 % | 12 % | 42 % |
> | **TRAP**                            | **100 %** | **100 %** | **100 %** | **99 %** | **63 %** | **94 %** |
>
> &nbsp;
>
> 7. > "Since TRAP steers images toward embedding alignment with high-level prompts, it could unintentionally amplify pre-existing biases in the CLIP model."
>     - **Response:** We agree with the reviewer that TRAP may amplify existing biases in the underlying VLM or CLIP model. Our intention with TRAP is to systematically expose such vulnerabilities, including ones that could lead to larger problems such as bias amplification, to motivate the development of stronger, semantically-aware defenses and more robust model architectures. We will clarify this point in the discussion section to highlight both the risks and the necessity of addressing them in future work.
> &nbsp;
>
> [1] Wang et al. “CogVLM: Visual Expert for Pretrained Language Models” (2024).

---

> ### Author Response · Authors · 2025-08-07
> **Follow-up and Final Clarifications**
>
> Thank you again for your thoughtful review and constructive suggestions. We have addressed all of your questions in detail in our rebuttal, including new results on updated models, additional e-commerce settings, hyperparameter sensitivity, robustness against state of the art defenses, and potential effects of our work on bias amplification. If you have any further concerns or would like additional clarification on any aspect, please let us know, we would be happy to elaborate further. We appreciate your time and consideration during the review process.

---

> > ### Comment · Reviewer_qNmy · 2025-08-08
> >
> > Thank you for adding more results and addressing all the raised points. I think the author addressed most my concerns. I encourage authors to add these results in the final version.

---

> > > ### Author Response · Authors · 2025-08-08
> > >
> > > We sincerely appreciate your comments on our work! We will make sure to add these additional results to the revised version of our paper.

---

### Official Review · Reviewer_Ce3w · 2025-07-14

**Clarity:** 3
**Significance:** 3
**Originality:** 3
**Rating:** 5
**Confidence:** 5

**Summary:**

This paper presents TRAP, an adversarial attack framework for vision-language model (VLM). The attack method creates adversarial image from text that affects VLM decision-making in a black box setting. Extensive experiments show the method achieve 100% attack success rate across multimodal models, and is robust to prompt/sampling noise. The black-box attack method along with its strong performance and transferability bring up a significant safety issue around VLM that are trained in a contrastive fashion.

**Questions:**

1. line 188: I am confused about the modulation process, if using mean(A), seems entire image embedding will be scale equivalently? How would this ensure semantic edits are concentrated in regions most relevant as indicated by mask? (line 190)
2. Can you elaborate a little on how TRAP's computational intensity look like, for example, what is the speed for generating an adversarial attacked image?
3. line 242, can you elaborate on the simple noise-based defense, is it just adding random noise to the adversarial attacked image?

**Ethical Concerns:**

["NO or VERY MINOR ethics concerns only"]

**Final Justification:**

Authors have provided results addressing my questions, and additionally addressed my concerns about the work (see my weakness argument). Therefore, I recommend acceptance to this work.

**Limitations:**

yes

**Quality:**

3

**Strengths And Weaknesses:**

Strength:
1. Clearly written, claims are well supported by method description and experiment.
2. High attack success rate and transferability across VLMs.
3. Robustness to simple prompt variation and decoding temperature.
4. Black box attack with no need to access VLM model weights.
5. Broader Impacts are discussed.

Weakness:
1. The attack is mainly for selection setting.
2. The attack depends highly on how VLM is trained and the performance of image generation model.

---

> ### Author Rebuttal · Authors · 2025-07-31
>
> Thank you for your valuable suggestions and feedback! We would like to address your concerns:
>
> 1. > “line 188: I am confused about the modulation process, if using mean(A), seems entire image embedding will be scale equivalently? How would this ensure semantic edits are concentrated in regions most relevant as indicated by mask? (line 190)”
>
>    - **Response:** Thank you for the suggestion. We originally used the mean-based modulation for computational simplicity and efficiency. After implementing the element-wise modulation, we observed no notable change in attack success rate or perceptual quality, but the runtime increased by approximately 18%. We will make this outcome and rationale clearer in the final version of the paper.
> &nbsp;
>
> 2. > “Can you elaborate a little on how TRAP's computational intensity look like, for example, what is the speed for generating an adversarial attacked image?”
>
>    - **Response:** We thank the reviewer for highlighting the importance of computational efficiency. As shown in Table 1, TRAP generates a single optimized adversarial image in an average of 666 seconds on an NVIDIA A100 GPU. The total time per sample for TRAP reflects the full iteration budget (20 steps × 20 iterations), representing a conservative, worst-case upper bound. In practice, TRAP’s early stopping mechanism, which is triggered when the attack success condition is met (i.e., the target model selects the adversarial image over the original in the selection setting), means most attacks succeed in fewer iterations, resulting in a **lower average runtime of 233.3 seconds** for the majority of samples.
> &nbsp;
>
> **Table&nbsp;1: Total Computation Time per Sample**
> | Method                          |    Step&nbsp;&nbsp;&nbsp;&nbsp; / Second |    Steps / Iteration&nbsp;&nbsp;&nbsp;&nbsp;    |    Iterations&nbsp;&nbsp;&nbsp;&nbsp;    |    Total Time (s) / Sample    |
> |-----------------------------|:------------------------:|:------------------:|:-----------:|:--------------------:|
> | **SPSA**                            | 0.94    | 20     | 20 | 168 s  |
> | **Bandit**                          | 0.00022 | 10 000 | 50 | 11 s   |
> | **Stable Diffusion (no opt.)**      | 4.6    | 1      | 1  | 4.6 s  |
> | **TRAP**                            | 0.6   | 20     | 20 | 666.6 s|
> &nbsp;
>
> 3. > “line 242, can you elaborate on the simple noise-based defense, is it just adding random noise to the adversarial attacked image?”
>
>    - **Response:** The simple noise-based defense involves adding Gaussian noise to the adversarial image, as in Byun et al. (2021)[1], to disrupt pixel-level attack gradients. However, since TRAP operates at the semantic and embedding level, its effectiveness is minimally affected by such noise. Attack success rate dropped by at most 1% (see Table 3). We also evaluated TRAP against CIDER[5] and MirrorCheck[6], which aim to mitigate both low-level and some higher-level perturbations, but TRAP consistently maintained high success rates. Finally, TRAP was tested on Robust-LLaVA[7], a model adversarially trained for robustness, and still achieved a 92% attack success rate. These results highlight that existing input-level and even adversarially trained defenses are insufficient to counter semantic attacks like TRAP.
> &nbsp;
>
> **Table&nbsp;2 · Robustness of TRAP Attack Under Different Defense Mechanisms and Adversarial Training**
> | COCO                               | &nbsp;LLaVA-1.5-34B&nbsp; | &nbsp;Gemma3-8B&nbsp; | &nbsp;Mistral-small-3.1-24B&nbsp; | &nbsp;Mistral-small-3.2-24B&nbsp; | &nbsp;Robust-LLaVA&nbsp; |
> |-------------------------------------------|:--------------:|:----------:|:--------------------------:|:--------------------------:|:-------------:|
> | **TRAP**                                  | 100 % | 100 % | 100 % | 97 % | 92 % |
> | **TRAP&nbsp;+Gaussian&nbsp;Noise (σ = 8/255)** | 100 % | 100 % | 100 % | 96 % | 92 % |
> | **TRAP&nbsp;+CIDER**                  | 100 % | 100 % | 96 %  | 90 % | 85 % |
> | **TRAP&nbsp;+MirrorCheck**            | 100 % | 98 %  | 88 %  | 82 % | 74 % |
>
> &nbsp;
>
> 4. > “The attack is mainly for selection setting.”
>
>    - **Response:** While our attack is primarily evaluated in the selection setting, this scenario is critical for real-world agentic GUI systems, where decisions often involve choosing the most relevant image or response from a set. The attack success rate in this context directly measures an AI agent’s vulnerability to manipulation in retrieval, moderation, or recommendation workflows. Vision-language models, such as those targeted in this paper, are the main decision-making module within the agentic systems. Thus, evaluating attacks on these multimodal models provides a practical and realistic surrogate for agentic workflows, as their architecture and selection logic closely match those in recent GUI Agent research, such as OS-ATLAS[2] and Aguvis[3].
> &nbsp;
>
> 5. > “The attack depends highly on how VLM is trained and the performance of image generation model.”
>
>    - **Response:** To address this, we evaluated TRAP on CogVLM[4], a model not trained with contrastive objectives, and found that TRAP still achieves strong attack success rates where other attacks fail (Table 3). Additionally, we ablated across multiple Stable Diffusion models and observed that TRAP’s effectiveness remains consistent regardless of the image generator used (Table 4).
> &nbsp;
>
> **Table 3 · Effectiveness of Adversarial Attacks on a Non-Contrastive Vision–Language Model (CogVLM)**
>
> | Model   | &nbsp;&nbsp;TRAP&nbsp;&nbsp; | &nbsp;&nbsp;SPSA&nbsp;&nbsp; | &nbsp;&nbsp;Bandit&nbsp;&nbsp; | &nbsp;&nbsp;Stable Diffusion (no opt.)&nbsp;&nbsp; | &nbsp;&nbsp;SSA_CWA&nbsp;&nbsp; | &nbsp;&nbsp;SA_AET&nbsp;&nbsp; |
> |:---------|:-----:|:-----:|:-------:|:---------------------------:|:--------:|:-------:|
> | **CogVLM** | **94 %** | 18 % | 0 % | 20 % | 4 % | 42 % |
> &nbsp;
>
> **Table 4 · Attack-Success Rates on Different Stable Diffusion Image Generators**
>
> | Model                        | LLaVA-1.5-34B | Gemma3-8B | Mistral-small-3.1-24B |
> |-------------------------|--------------:|----------:|----------------------:|
> | **stabilityai/stable-diffusion-2-1**            | 100 % | 100 % | 100 % |
> | **stabilityai/stable-diffusion-xl-base-1.0**    | 100 % | 100 % | 100 % |
> | **stable-diffusion-v1-5 / stable-diffusion-v1-5** | 100 % | 100 % | 100 % |
>
> **References**
>
> [1] Byun et al. “On the Effectiveness of Small Input Noise for Defending Against Query-based Black-Box Attacks” (2021).
>
> [2] Wu et al. “OS-ATLAS: A Foundation Action Model for Generalist GUI Agents” (2024).
>
> [3] Xu et al. “Aguvis: Unified Pure Vision Agents for Autonomous GUI Interaction ” (2024).
>
> [4] Wang et al. “CogVLM: Visual Expert for Pretrained Language Models” (2024).
>
> [5] Xu et al. "Cross-modality Information Check for Detecting Jailbreaking in Multimodal Large Language Models" (2024).
>
> [6] Fares et al. "MirrorCheck: Efficient Adversarial Defense for Vision-Language Models" (2024).
>
> [7] Malik et al. "Robust-LLaVA: On the Effectiveness of Large-Scale Robust Image Encoders for Multi-modal Large Language Models" (2025).

---

> > ### Comment · Reviewer_Ce3w · 2025-08-08
> >
> > Thank you for addressing my questions and providing speed analysis as well as additional experiments on different VLM and image generators. These additional results addressed my statements about weakness of this work. I recommend acceptance of this work.

---

> ### Author Response · Authors · 2025-08-07
> **Follow-up and Final Clarifications**
>
> Thank you again for your thorough review and thoughtful questions. We hope our responses in the rebuttal have fully addressed your concerns regarding the modulation process, computational intensity, defense mechanisms, and the scope of TRAP. If there are any remaining questions or further concerns, please let us know, we would be happy to clarify or provide additional details. We appreciate your time and consideration throughout the review process.

---

> ### Author Response · Authors · 2025-08-08
>
> As the author response period is closing soon, please let us know if there are any remaining concerns or clarifications you would like us to address. We appreciate your time and feedback, and we are happy to provide any additional details as needed.

---

### Decision · Program_Chairs · 2025-09-17

**Decision:**

Accept (poster)

**Comment:**

This paper proposes a black-box adversarial attack framework for vision-language models to manipulate their preferences, called TRAP. It optimizes CLIP embedding space with a combination of perceptual, semantic, and identity-preserving loss, and then generates a natural-looking image with the Stable Diffusion model. The empirical results demonstrate its effectiveness across multiple benchmarks and settings, achieving 100% ASR against current models.

The threat raised by this paper is important and timely. Its effectiveness across different scenarios can alarm the community to notice such vulnerabilities. The weakness is mainly that its attack method design is not novel and systemical enough and discussions on the possible defense is not sufficient. During the rebuttal, the authors provided additional clarifications on the attack design, explained the practical relevance of their scenarios, etc. These responses address the main concerns of all the reviewers, and both of them pose a positive attitude toward this paper.

Overall, I find the work proposes an important and impactful threat with solid evaluations. Therefore, I suggest accepting the paper as a poster. The reason why I do not recommend it as a spotlight/oral is its method novelty and limited discussions on the possible defense.